# BRIDGING THE PERFORMANCE-GAP BETWEEN TARGET-FREE AND TARGET-BASED REINFORCEMENT LEARNING

**Théo Vincent**[1,2,*]  **Yogesh Tripathi**[1,2]  **Tim Faust**[1,2]  **Abdullah Akgül**[3]
**Yaniv Oren**[5]  **Melih Kandemir**[3]  **Jan Peters**[1,2,4,7]  **Carlo D'Eramo**[6]

[1]DFKI, SAIROL  [2] TU Darmstadt  [3] University of Southern Denmark  [4] Hessian.ai
[5] Delft University of Technology  [6] University of Würzburg  [7] Robotics Institute Germany

## ABSTRACT

The use of target networks in deep reinforcement learning is a widely popular solution to mitigate the brittleness of semi-gradient approaches and stabilize learning. However, target networks notoriously require additional memory and delay the propagation of Bellman updates compared to an ideal target-free approach. In this work, we step out of the binary choice between target-free and target-based algorithms. We introduce a new method that uses a copy of the last linear layer of the online network as a target network, while sharing the remaining parameters with the up-to-date online network. This simple modification enables us to keep the target-free's low-memory footprint while leveraging the target-based literature. We find that combining our approach with the concept of iterated $Q$-learning, which consists of learning consecutive Bellman updates in parallel, helps improve the sample-efficiency of target-free approaches. Our proposed method, iterated Shared $Q$-Learning (iS-QL), bridges the performance gap between target-free and target-based approaches across various problems while using a single $Q$-network, thus stepping towards resource-efficient reinforcement learning algorithms.

## 1 INTRODUCTION

Originally, $Q$-learning (Watkins & Dayan, 1992) was introduced as a reinforcement learning (RL) method that performs asynchronous dynamic programming using a single look-up table. By storing only one $Q$-estimate, $Q$-learning benefits from an up-to-date estimate and a low memory footprint. However, replacing look-up tables with non-linear function approximators and allowing off-policy samples to make the method more scalable introduces training instabilities (Sutton & Barto, 2018). To address this, Mnih et al. (2015) introduce Deep $Q$-Network (DQN), an algorithm that constructs the regression target from an older version of the online network, known as the *target network*, which is periodically updated to match the online network (see "Target Based" in Figure 1). This modification to the temporal-difference objective helps mitigate the negative effects of function approximation and bootstrapping (Zhang et al., 2021), two elements of the deadly triad (van Hasselt et al., 2018). Recently, new methods have demonstrated that increasing the size of the $Q$-network can enhance the learning speed and final performance of temporal difference methods (Espeholt et al., 2018; Schwarzer et al., 2023; Nauman et al., 2024; Lee et al., 2025). Numerous ablation studies highlight the crucial role of the target network in maintaining performance improvements over smaller networks (Figure 7 in Schwarzer et al. (2023), and Figure 9$b$ in Nauman et al. (2024)). Interestingly, even methods initially introduced without a target network (Bhatt et al. (2024) and Kim et al. (2019)) benefit from its reintegration (Figure 5 in Palenicek et al. (2025) and Gan et al. (2021)).

While temporal difference methods clearly benefit from target networks, their utilization doubles the memory footprint dedicated to $Q$-networks. This ultimately limits the size of the online network due to the constrained memory capacity of the processing units (e.g. the vRAM of a GPU). This limitation is not only problematic for learning on edge devices where memory is constrained, but also for applications that inherently require large network sizes, such as handling high-dimensional state spaces (Boukas et al., 2021; Pérez-Dattari et al., 2019), processing multi-modal inputs (Schneider et al., 2026), or constructing mixtures of experts (Obando Ceron et al., 2024; Hendawy et al., 2024). This motivates the development of target-free methods (see "Target Free" in Figure 1).

---

*Correspondance to: `theo.vincent@dfki.de`
Code available at: `https://github.com/theovincent/iS-DQN`

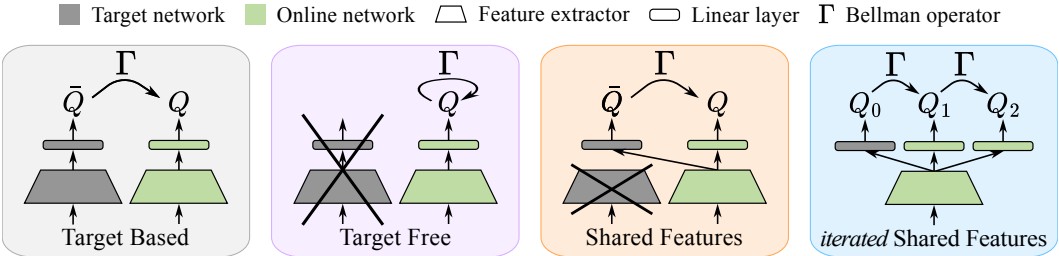

Figure 1: We propose a simple alternative to target-based/target-free approaches, where a linear layer represents the target network, sharing the rest of the parameters with the online network (Shared Features). We apply the concept of iterated $Q$-learning (Vincent et al., 2025b), which consists of learning multiple Bellman updates in parallel, to reduce the performance gap between target-free and target-based approaches (*iterated* Shared Features).

In this work, we introduce an alternative to the binary choice between target-free and target-based approaches. We propose storing only the parameters of the last linear layer, while using the parameters of the online network to substitute the other layers of the target network (see "Shared Features" in Figure 1). Although this simple modification alone helps reduce the performance gap between target-free and target-based DQN (see "iS-DQN $K = 1$" in Figure 4, right), we explain in this work how it opens up the possibility of leveraging the target-based literature to reduce this gap further, while maintaining a low memory footprint. Notably, this approach is also orthogonal to regularization techniques that have been shown to be effective for target-free algorithms (Kim et al., 2019; Bhatt et al., 2024; Gallici et al., 2025). Therefore, we will build upon these approaches to benefit from their performance gains.

In the following, we leverage the concept of iterated $Q$-learning (Vincent et al., 2025b) to enhance the learning speed (in terms of the number of environment interactions) of target-free algorithms, which is a major bottleneck in many real-world applications. The concept of iterated $Q$-learning, initially introduced as a target-based approach, aims at learning multiple Bellman iterations in parallel. This leads to a new algorithm, termed iterated Shared $Q$-Network (iS-QN), pronounced "ice-QN" to emphasize that it contains a frozen head. iS-QN utilizes a single network with multiple linear heads, where each head is trained to represent the Bellman target of the previous one (see "*iterated* Shared Features" in Figure 1). Our evaluation of iS-QN across various RL settings demonstrates that it improves the learning speed of target-free methods while maintaining a comparable memory footprint and training time.

## 2 BACKGROUND

**Deep $Q$-Network (Mnih et al., 2015)**   The optimal policy of a Markov Decision Process (MDP) can be obtained by selecting for each state, the action that maximizes the optimal action-value function $Q^*$. This function represents the largest achievable expected sum of discounted rewards given a state-action pair. In the context of discrete action spaces, Mnih et al. (2015) approximate the optimal action-value function with a neural network $Q_\theta$, represented by a vector of parameters $\theta$. This neural network is learned to approximate its Bellman iteration $\Gamma Q_\theta$, leveraging the contraction property of the Bellman operator $\Gamma$ in the value space to guide the optimization process toward the operator's fixed point, i.e., the optimal action-value function $Q^*$. In practice, a sample estimate of the Bellman iteration is used, where for a sample $(s, a, r, s')$, $\Gamma Q_\theta(s, a) = r + \gamma \max_{a'} Q_\theta(s', a')$, where $\gamma$ is the discount factor. However, this learning procedure is unstable because the neural network $Q_\theta$ learns from its own values, which change at each optimization step due to function approximation, and because of the compound effect of the overestimation bias. To mitigate these issues, the authors introduce a target network with parameters $\bar{\theta}$ to stabilize the regression target $\Gamma Q_{\bar{\theta}}$, and periodically update these parameters to the online parameters $\theta$ every $T$ steps. On the negative side, this doubles the memory footprint dedicated to $Q$-networks.

**Iterated $Q$-Network (Vincent et al., 2025b)**   By using a target network, DQN slows down the training process as multiple gradient steps are dedicated to each Bellman iteration, as $\Gamma Q_{\bar{\theta}}$ is delayed by some gradient steps compared to $\Gamma Q_\theta$. To increase the learning speed, Vincent et al. (2025b) propose to learn consecutive Bellman iterations in parallel. This approach uses a sequence of online parameters $(\theta_i)_{i=1}^{K}$ and a sequence of target parameters $(\bar{\theta}_i)_{i=0}^{K-1}$. Each online network $Q_{\theta_{i+1}}$ is

trained to regress $\Gamma Q_{\bar{\theta}_i}$. Similarly to DQN, each target parameter $\bar{\theta}_i$ is updated to the online parameter $\theta_{i+1}$ every $T$ steps. Importantly, the structure of a chain is enforced by setting each $\bar{\theta}_i$ to $\theta_i$ every $D \ll T$ steps so that each $Q_{\theta_{i+1}}$, which is learned to regress $\Gamma Q_{\bar{\theta}_i}$, are forced to approximate $\Gamma Q_{\theta_i}$. This results in $Q_{\theta_K} \approx \Gamma Q_{\theta_{K-1}} \approx \ldots \approx \Gamma^K Q_{\theta_0}$, thus learning $K$ consecutive Bellman iterations in parallel. Importantly, DQN can be recovered by setting $K = 1$. While the feature representation can be shared across the online $Q$-networks, iterated $Q$-Network (i-QN) has the drawback of requiring an old copy of the online networks to stabilize training, significantly increasing the memory footprint. In the following, we will explain how the concept of i-QN can help reduce the performance gap between target-free and target-based approaches while maintaining a low memory footprint.

## 3 RELATED WORK

Other works have considered removing the target network in different RL scenarios. Vasan et al. (2024) introduce Action Value Gradient, an algorithm designed to work well in a streaming scenario where no replay buffer, no batch updates, and no target networks are available. Gallici et al. (2025) also develop a method for a streaming scenario, in which they rely on parallel environments to cope with the non-stationarity of the sample distribution. Gradient Temporal Difference learning is another line of work that does not use target networks (Sutton et al., 2009; Maei et al., 2009; Yang et al., 2021; Patterson et al., 2022; Elelimy et al., 2025). Instead, they compute the gradient w.r.t. the regression target as well as the gradient w.r.t. the predictions, which doubles the compute requirement. Additionally, to address the double sampling problem, another network is trained to approximate the temporal difference value, which also increases the memory footprint.

Alternatively, some works construct the regression target from the online network instead of the target network, but still use a target network in some other way. For example, Ohnishi et al. (2019) compute the TD(0) loss from the online network and add a term in the loss to constrain the predictions of the online network for the next state-action pair $(s', a')$ to remain close to the one predicted by the target network. Piché et al. (2021; 2023) develop a similar approach, enforcing similar values for the state-action pair $(s, a)$. Lindström et al. (2025) show that the target network can be removed after a pretraining phase in which they rely on expert demonstrations. Hendawy et al. (2026) construct the target by using the lowest estimate between the online and target networks' predictions to learn from up-to-date estimates without suffering from the overestimation bias.

Many regularization techniques have been developed, attempting to combat the performance drop that occurs when removing the target network. We stress that our approach is orthogonal to these regularization techniques and we show in Section 5 that our method improves the performance of target-free methods equipped with these advancements. Li & Pathak (2021) encode the input of the $Q$-network with learned Fourier features. While this approach seems promising, the authors acknowledge that the performance degrades for high-dimensional problems. Shao et al. (2022) remove the target-network and search for an action that maximizes the $Q$-network predictions more than the action proposed by the policy. Searching for a better action requires additional resources and is only relevant for actor-critic algorithms. Kim et al. (2019) leverage the MellowMax operator to get rid of the target network. However, the temperature parameter needs to be tuned (Kim, 2020), which increases the compute budget, and a follow-up work demonstrates that the reintegration of the target network is beneficial (Gan et al., 2021). We combine the MellowMax operator with the presented approach in Section D.1 and demonstrate that it bridges the performance gap between the target-free and target-based approaches. Finally, Bhatt et al. (2024) point out the importance of using batch normalization (Ioffe & Szegedy, 2015) to address the distribution shift of the input given to the critic. Our investigation reveals that it degrades the performance in a discrete action setting (see Figure 21, right).

The idea of learning multiple Bellman iterations has been introduced by Schmitt et al. (2022). They demonstrate convergence guarantees in the case of linear function approximation. Then, Vincent et al. (2024) used this approach to learn a recurrent hypernetwork generating a sequence of $Q$-functions where each $Q$-function approximates the Bellman iteration of the previous $Q$-function. Finally, Vincent et al. (2025b) introduced iterated $Q$-Network as a far-sighted version of DQN that learns the $K$ following Bellman iterations in parallel instead of only learning the following one. While promising, those approaches rely on a separate copy of the learnable parameters to stabilise the training process, which increases the memory footprint. In this work, we propose to leverage the potential of iterated $Q$-learning to boost the learning speed of target-free algorithms.

---

**Algorithm 1** *iterated* Shared Deep $Q$-Network (iS-DQN). Modifications to DQN are in purple.

---

1: Initialize a network $Q_\theta$ with $K+1$ heads, where each head is defined by the parameters $\omega_k$. We note $\theta_k = (\omega, \omega_k)$, and $\omega$ the shared parameters such that $\theta = (\omega, \omega_0, .., \omega_K)$. $\mathcal{D}$ is an empty replay buffer.
2: **Repeat**
3:      Set $u \sim \text{Uniform}(\{1, .., K\})$.
4:      Take action $a \sim \epsilon\text{-greedy.}(Q_{\theta_u}(s, \cdot))$; Observe reward $r$, next state $s'$.
5:      Update $\mathcal{D} \leftarrow \mathcal{D} \bigcup \{(s, a, r, s')\}$.
6:      **every $G$ steps**
7:          Sample a mini-batch $\mathcal{B} = \{(s, a, r, s')\}$ from $\mathcal{D}$.
8:          Store $[Q_0(s', \cdot), .., Q_K(s', \cdot)] \leftarrow Q_\theta(s', \cdot)$ and $[Q_0(s, a), .., Q_K(s, a)] \leftarrow Q_\theta(s, a)$.
9:          Compute the loss                  $\triangleright \lceil \cdot \rceil$ indicates a stop gradient operation.
         $\mathcal{L}^{\text{iS-QN}} = \sum_{(s,a,r,s') \in \mathcal{B}} \sum_{k=1}^{K} (\lceil r + \gamma \max_{a'} Q_{k-1}(s', a') \rceil - Q_k(s, a))^2$.
10:          Update $\theta$ from $\nabla_\theta \mathcal{L}^{\text{iS-QN}}$.
11:      **every $T$ steps**
12:          Update $\omega_k \leftarrow \omega_{k+1}$, for $k \in \{0, \ldots, K-1\}$.

---

# 4 METHOD

Our goal is to design a new algorithm that improves the learning speed of target-free value-based RL methods without significantly increasing the number of parameters used by the $Q$-networks. To achieve this, we consider a *single* $Q$-network parameterized with $K+1$ heads. We note $\omega_k$ the parameters of the $k^{\text{th}}$ head, $\omega$ the shared parameters, and define $\theta = (\omega, \omega_0, .., \omega_K)$ and $\theta_k = (\omega, \omega_k)$. Following Vincent et al. (2025b), for a sample $d = (s, a, r, s')$, the training loss is

$$\mathcal{L}_d^{\text{iS-QN}}(\theta) = \sum_{k=1}^{K} \mathcal{L}_d^{\text{QN}}(\theta_k, \theta_{k-1}), \tag{1}$$

where $\mathcal{L}_d^{\text{QN}}$ can be chosen from any temporal-difference learning algorithm. For instance, DQN uses $\mathcal{L}_d^{\text{QN}}(\theta_k, \theta_{k-1}) = (\lceil r + \gamma \max_{a'} Q_{\theta_{k-1}}(s', a') \rceil - Q_{\theta_k}(s, a))^2$, where $\lceil \cdot \rceil$ indicates a stop gradient operation. We stress that $\omega_0$ is not learned. However, every $T$ steps, each $\omega_k$ is updated to $\omega_{k+1}$, similarly to the target update step in DQN. This way, iS-QN allows to learn $K$ Bellman iterations in parallel while only requiring a small amount of additional parameters on top of a target-free approach. Indeed, in the general case, the size of each head $\omega_k$ is negligible compared to the size of shared parameters $\omega$. Algorithm 1 summarizes the changes brought to the pseudo-code of DQN to implement this approach.

In Figure 2, we compare the training paths defined by the $Q$-functions obtained after each target update of the proposed approach (top) and the target-based approach (bottom). For each given sample, the target-based approach learns only 1 Bellman iteration at a time and proceeds to the following one after $T$ training steps. In contrast, the iterated Shared Features approach learns several consecutive Bellman iterations in parallel for each given sample. The considered window also moves forward every $T$ training steps. As the window shifts, the network represents $Q$-functions that are closer to the optimal $Q$-function since every $Q$-function is learned to represent the Bellman iteration of the previous $Q$-function. Similarly to the target-based and target-free approaches, the online parameters are updated with the gradient computed through the forward pass of the state-action pair $(s, a)$, as indicated with blue arrows. In Figure 2, we depict our approach with $K = 2$. However, the number of heads can be increased at minimal cost. We note that the first $Q$-function is considered fixed in this representation, even if the head is the only frozen element and the previous layers are shared with the other learned $Q$-estimates. We remark that iS-QN with $K = 1$ implements the "Shared Features" approach presented in Figure 1. Interestingly, the target-free approach can also be depicted in Figure 2. Indeed, not using a target network is equivalent to updating the target network to the online network after each gradient step. Consequently, the target-free approach can be understood as the target-based representation with a window shifting at every step. Therefore, the target-free approach passes through the Bellman iterations faster, creating instabilities as the optimization landscape may direct the training path toward undesirable $Q$-functions.

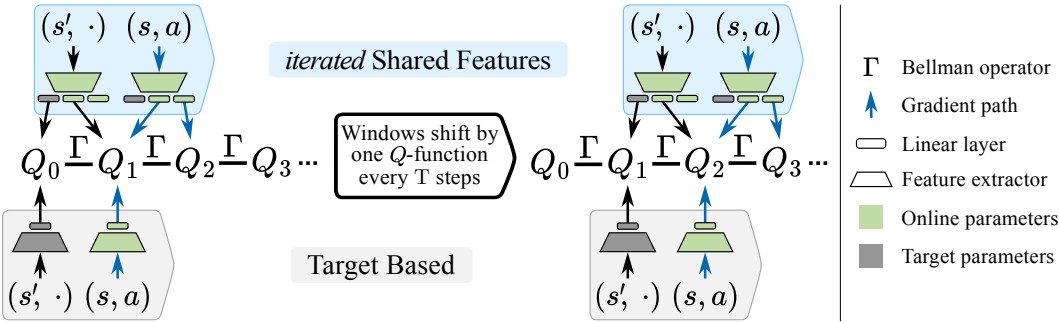

Figure 2: Comparison of the training path defined by the target networks obtained after each target update during training between the target-based approach (bottom) and the *iterated* Shared Features approach (top). While both approaches wait for $T$ training steps before shifting their respective window by one $Q$-function, our approach already considers the following Bellman iterations using multiple heads, where each head represents the Bellman iteration of the previous head.

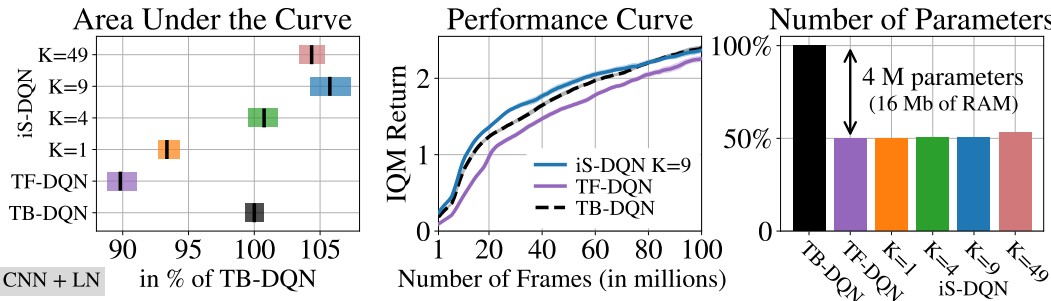

Figure 3: Reducing the performance gap in online RL on 15 **Atari** games with the CNN architecture and LayerNorm (LN). While removing the target network leads to a $10\%$ drop in AUC (left), iS-DQN $K = 9$ (using $10$ linear heads), not only closes the gap but improves over the target-based approach by $6\%$. Importantly, iS-DQN uses a comparable number of parameters to TF-DQN (right).

In the following, we apply iterated Shared Features to several target-based approaches on multiple RL settings, demonstrating that it reduces the gap between target-free and target-based methods. For each algorithm A, we note TB-A as its target-based version, TF-A as its target-free version, and iS-A as the iterated shared approach, where "iS" stands for iterated Shared. For example, for the experiments with the DQN algorithm, we note TF-DQN, the target-free version, and iS-DQN, the iterated Shared version. Importantly, we incorporate the insights provided by Gallici et al. (2025) to use LayerNorm (Ba et al., 2016) for the experiments with discrete action spaces, as we found it beneficial, even for the target-based approach. Similarly, we use BatchNorm (Ioffe & Szegedy, 2015), as suggested by Bhatt et al. (2024), to improve sample-efficiency in continuous action settings, except for the target-based approach, as it degrades performances (see Figure 25, right).

## 5 EXPERIMENTS

We evaluate iS-QN in online, offline, continuous control, and language-based RL scenarios to demonstrate that it can enhance the learning speed of target-free methods. We focus on the learning speed because, in this work, we are interested in the sample efficiency of target-free methods. We use the Area Under the performance Curve (AUC) to measure the learning speed. The AUC has the benefit of depending less on the training length compared to the end performance, as it accounts for the performance during the entire training. It also favors algorithms that constantly improve during training over those that only emerge at the end of training, thus penalizing algorithms that require many samples to perform well. In each experiment, we report the AUC of each algorithm, normalized by the AUC of the target-based approach, to facilitate comparison. By normalizing the AUCs, the resulting metric can also be interpreted as the average performance gap observed during

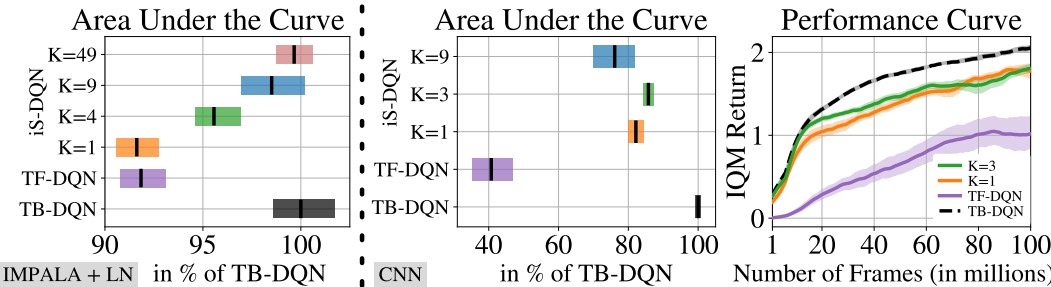

Figure 4: **Left**: Reducing the performance gap in online RL on 10 **Atari** games with the IMPALA architecture and LayerNorm (LN). Similar to the results with the CNN architecture, iS-DQN bridges the gap between the target-free and target-based approaches. **Middle** and **Right**: Reducing the performance gap in online RL on 15 **Atari** games with the CNN architecture. Removing the target network of the vanilla DQN algorithm results in a $60\%$ performance drop ($100\% - 40\%$). By using iS-DQN with $K = 3$, the performance drop is divided by 4 ($100\% - 85\% = 15\% = 60\%/4$), thereby confirming the benefit of this approach.

training between the considered approach and the target-based approach. We use the Inter-Quantile Mean (IQM) and $95\%$ stratified bootstrapped confidence intervals to allow for more robust statistics as advocated by Agarwal et al. (2021). The IQMs are computed over 5 seeds per Atari game, 10 seeds per DMC Hard tasks, and 5 seeds for Wordle. 15 Atari games are used for the experiments on the CNN architecture, and 10 games for the experiments on the IMPALA architecture to reduce the computational budget. Importantly, all hyperparameters are kept untouched with respect to the standard values (Castro et al., 2018), only the architecture is modified, as described in Section 4. Extensive details about the selection process of the Atari games, the metrics computation, the hyperparameters, and the individual learning curves are reported in the appendix.

## 5.1 ONLINE DISCRETE CONTROL

First, we evaluate iS-DQN on 15 Atari games (Bellemare et al., 2013) with the vanilla CNN architecture (Mnih et al., 2015) equipped with LayerNorm. As expected, the target-free approach yields an AUC $10\%$ smaller than the target-based approach, as shown in Figure 3 (left). This performance drop is constant across the training, see Figure 3 (middle). Interestingly, iS-DQN $K = 1$ improves over TF-DQN by simply storing an old copy of the last linear head. As more Bellman iterations are learned in parallel, the performance gap between iS-DQN and TB-DQN shrinks. Remarkably, iS-DQN $K = 9$ even outperforms the target-based approach by $6\%$ in AUC. We note a slight decline in performance for iS-DQN $K = 49$. We conjecture that this is due to the shared feature representation not being rich enough to enable the network to learn 49 Bellman iterations in parallel with linear approximations. Importantly, Figure 3 (right) testifies that this performance boost is achieved with approximately half of the parameters used by the target-based approach, truly reducing the memory footprint required by the $Q$-functions.

Our evaluation with the IMPALA architecture (Espeholt et al., 2018) with LayerNorm confirms the ability of iS-DQN to reduce the performance gap between target-free and target-based approaches. Indeed, Figure 4 (left) indicates that removing the target network leads to an $8\%$ performance drop while iS-DQN annuls the performance gap as more Bellman iterations are learned in parallel, i.e., as $K$ increases. Interestingly, as opposed to the CNN architecture, increasing the number of heads to learn 49 Bellman iterations in parallel is beneficial in this scenario. We believe this is due to IMPALA architecture's ability to produce a richer representation than the CNN architecture, thereby allowing more Bellman iterations to be approximated with a linear mapping. The plots of the performance curve and the number of parameters are similar to the ones for the CNN architecture, see Figure 19.

Finally, we confirm the benefit of the iterated Shared Features approach by removing the normalization layers for all algorithms with the CNN architecture in Figure 4 (right). We observe a major drop in performance for TF-DQN, leading to $60\%$ performance gap ($100\% - 40\%$). Notably, iS-DQN $K = 1$ reduces this performance gap to $18\%$ ($100\% - 82\%$). This highlights the potential of simply storing the last linear layer and using the features of the online network to build a lightweight regression

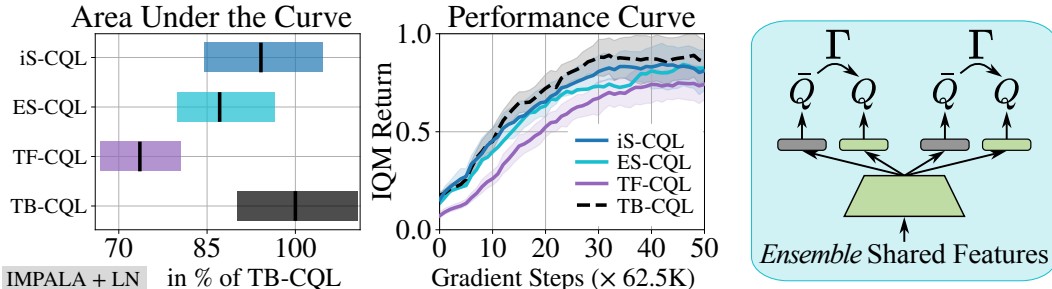

Figure 5: Reducing the performance gap in offline RL on 10 **Atari** games with the IMPALA architecture and LayerNorm (LN). iS-CQL shrinks the performance gap from $26\%$ to $6\%$. Interestingly, applying the idea of sharing parameters to Ensemble DQN (*Ensemble* Shared Features, ES-CQL) also reduces the performance gap, demonstrating that this idea is not limited to iterated $Q$-learning and can be applied to other target-based approaches.

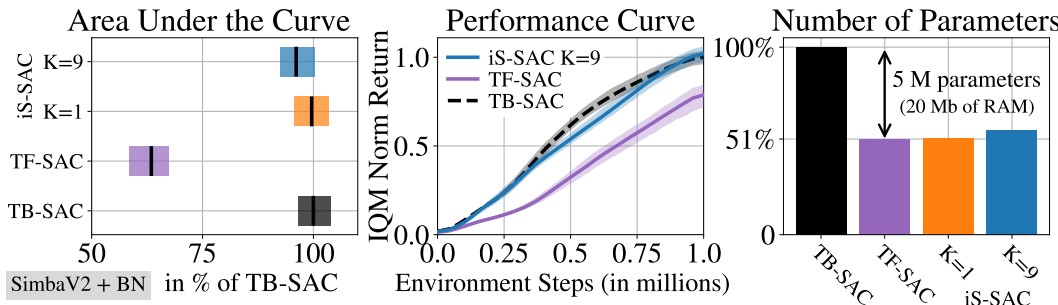

Figure 6: Reducing the performance gap in online RL on the 7 **DMC Hard** tasks with the SimbaV2 architecture and BatchNorm (BN). iS-SAC recovers the performance drop incurred by removing the target network (left). This performance boost is made while reducing the *total* number of parameters by $49\%$ (right).

target. While increasing the number of learned Bellman iterations to 3 brings a small benefit, the performances are slightly decreasing for higher values of $K$, indicating that LayerNorm is beneficial to provide useful representations when considering a higher number of linear heads.

## 5.2 OFFLINE DISCRETE CONTROL

We consider an offline RL setting in which the agent has access to $10\%$ of the dataset collected by a vanilla DQN agent trained with a budget of 200 million frames (Agarwal et al., 2020), sampled uniformly. We adapt the loss for learning each Bellman iteration to the one proposed by Kumar et al. (2020b). This leads to an iterated version of Conservative $Q$-Learning (CQL). In Figure 5, iS-CQL $K = 9$ reduces the performance gap by 20 percentage points, ending up with a performance gap of $6\%$ compared to $26\%$ for TF-CQL. Additionally, we evaluate another way of sharing features to show that this idea is not limited to iterated $Q$-learning. Instead of building a chain of $Q$-functions represented by linear heads, we define an ensemble of pairs of linear heads. Each pair contains a frozen head representing a target network $\bar{Q}$ that is used to train the learned head representing the associated online network $Q$, as depicted in Figure 5 (right). We evaluate this variant that we call Ensemble Shared Features (ES-CQL), with 5 pairs of heads, i.e. 10 heads, to match the number of heads used by iS-CQL $K = 9$, as the number of heads of iS-QN is always equal to $K + 1$. Importantly, ES-CQL also outperforms TF-CQL, reinforcing the idea that sharing parameters and using linear heads is a fruitful direction.

## 5.3 ONLINE CONTINUOUS CONTROL

We investigate the behavior of iS-QN on the DeepMind Control suite (Tassa et al., 2018), focusing on the hard tasks. We select Soft Actor-Critic (SAC, Haarnoja et al. (2018)) as the base algorithm and

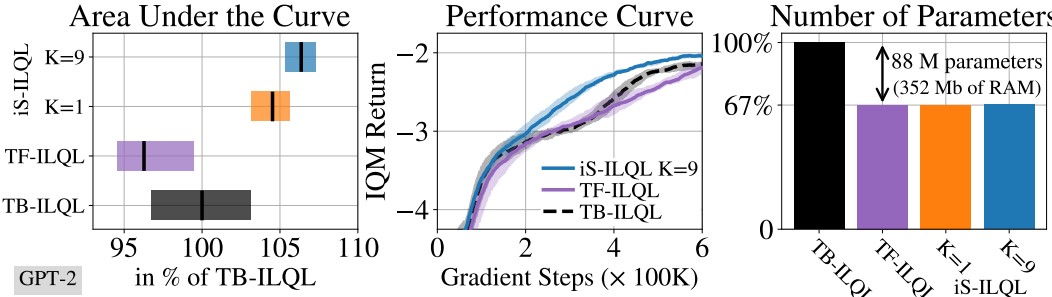

Figure 7: Reducing the performance gap in offline RL on **Wordle** with the GPT-2 small architecture. iS-ILQL $K = 9$, not only closes the gap but improves over the target-based approach by more than $5\%$. Importantly, iS-ILQL saves $33\%$ of RAM compared to the target-based approach (right).

adapt the architecture to the one proposed by Lee et al. (2025) (SimbaV2) so that the target-based approach corresponds to the state-of-the-art. This experiment allows us to test iS-QN on different learning dynamics, as the target updates are done with an exponentially moving average instead of a hard update, and the loss for the critic uses a categorical distribution to learn the distribution of the return. Interestingly, Figure 6 (left) shows that only using an old copy of the last layer of the critic to construct the regression target (iS-SAC $K = 1$) recovers the performance drop incurred by the target-free approach compared to the target-based approach. Importantly, Lee et al. (2025) design the critic with significantly more parameters than the actor, as commonly done in the actor-critic literature (Mysore et al., 2021; Mastikhina et al., 2025). This means that iS-SAC K=1 reduces the *total* number of parameters by $49\%$, see Figure 6 (right). When considering more heads to learn the following Bellman updates, we find it beneficial to give more importance to the first Bellman updates by scaling the future terms in the loss by a discounting factor of $0.25$. We leave the investigation of finding the best way to weight each term in the loss to future work. In Section D.4, we provide a first direction leveraging the concept of meta-gradient reinforcement learning (Xu et al., 2018) to tune learnable coefficients assigned to each term in the loss during training. We note that in this setting, iS-SAC $K = 9$ only performs on par with $K = 1$. Nonetheless, iS-SAC $K = 9$ is still performing better than the best target-free approach, having overlapping confidence intervals with the target-based approach, which serves as the gold standard, as it requires additional parameters.

## 5.4 SCALING UP TO LANGUAGE MODELS

In this experiment, we evaluate iS-QN on an offline RL language processing task. Specifically, we focus on Implicit Language $Q$-Learning (ILQL, Snell et al. (2023)), a method introduced with a target network. It adapts implicit $Q$-learning (Kostrikov et al., 2023) to the language domain by sampling action tokens from a policy, learned with supervised learning, and weighted by the advantage computed from the $Q$-function. We evaluate ILQL on the Wordle game (Lokshtanov & Subercaseaux, 2022), a multi-turn game where the agent guesses a hidden word and receives feedback after each attempt. As in Snell et al. (2023), we choose the GPT-2 small architecture, which results in TB-ILQL using 264 million parameters. In Figure 7 (left), we note that while a performance drop is noticeable, the target-free approach does not perform significantly worse than the target-based variant. Importantly, sharing parameters and learning $K = 9$ Bellman iterations in parallel improves the learning speed of the target-free approach by $10\%$ without significantly increasing the memory footprint. This leads iS-QN to save 88 million parameters compared to the original approach.

## 5.5 STREAMING REINFORCEMENT LEARNING

We consider the streaming setting in which the RL agent only learns from a stream of data. In this setting, the agent does not have access to a replay buffer, batch updates, or parallel environments. These constraints position the RL agent in a drastically different setting from the previously considered ones, as replay buffers and batch updates greatly stabilize the learning process (Vasan et al., 2024). Elsayed et al. (2024) introduced Stream Q($\lambda$), an algorithm adapted for the streaming setting. It is an adaptation of the original Q($\lambda$) algorithm that uses observation and reward normalization, sparse initialization, layer norm, and an adaptive step-size optimizer. In this experiment, we combine

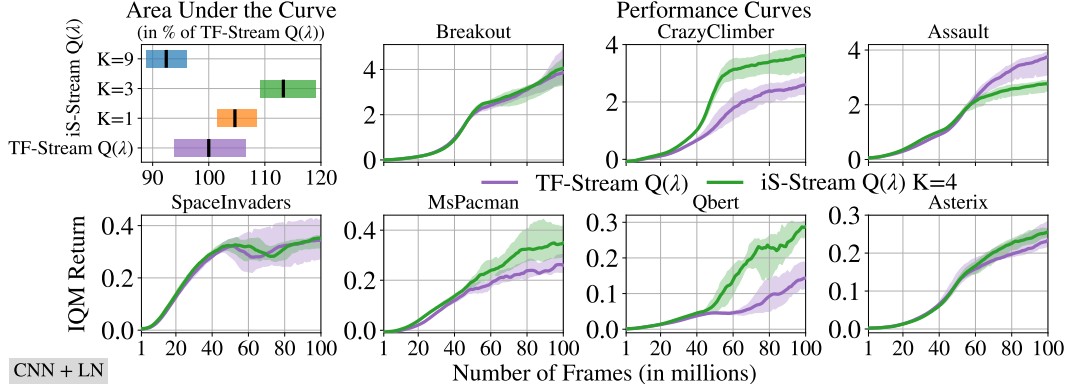

Figure 8: Increasing the learning speed in a streaming scenario on 7 **Atari** games with the CNN architecture and LayerNorm (LN). iS-Stream Q($\lambda$) $K = 3$ improves over the target-free approach by more than $10\%$, outperforming or performing on par with the baseline on 6 out of the 7 games.

Stream Q($\lambda$) with the presented approach to obtain iS-Stream Q($\lambda$). We remark that Stream Q($\lambda$) was introduced without a target network; therefore, we focus only on this version (TF-Stream Q($\lambda$)). We fix a target update period of $T = 10$ for iS-Stream Q($\lambda$). In Figure 8, we report the performance of the target-free approach and the presented approach for three values of $K$ across 7 Atari games that were selected for their diversity in human-normalized score (see Figure 10). iS-Stream Q($\lambda$) $K = 1$ performs similarly to the target-free variant. Remarkably, increasing the number of heads to 4 ($K = 3$) improves the learning speed compared to the target-free approach. In this setting, increasing the number of heads until 9 Bellman iterations are learned in parallel is not helpful. The absence of batch updates increases the variance of the gradient steps, potentially leading to unstable updates for the shared parameters for a large number of heads.

## 5.6 WHY IS IS-QN IMPROVING OVER TARGET-FREE APPROACHES?

We now provide some insights to understand why iS-QN reduces the performance gap between target-free and target-based approaches. First, we investigate the change in the learning dynamics that happens when the features are shared between the online and the target heads ("Shared Features" or equivalently, "*iterated* Shared Features" with $K = 1$, see Figure 1). To evaluate the impact on the learning dynamics, we compute, for each gradient step of an iS-DQN $K = 1$ agent, the gradient with respect to the loss of iS-DQN, as well as the gradients that the target-based loss and the target-free loss would produce. These quantities determine how the parameters evolve during training. We then report the cosine similarity between the gradients w.r.t. the iS-QN loss and the TB-DQN loss, and the cosine similarity between the gradients w.r.t. the TF-DQN loss and the TB-DQN loss in Figure 9 (left) for 15 Atari games. Interestingly, the gradients obtained by the target-based approach are closer to the gradients of iS-DQN $K = 1$ than the gradients of the target-free approach, especially at the beginning of the training. This means that by simply using a copy of the last linear layer and sharing features, iS-DQN's learning dynamics become closer to those of the target-based approach.

At first sight, the fact that iS-QN uses frozen heads on top of features changing at each gradient step might seem like an uncommon practice in machine learning. However, this design choice is already part of the reinforcement learning literature. Indeed, in Deep $Q$-Network, the $Q$-network is designed with multiple heads, each one representing the prediction for a specific action. For each sample, only the selected head corresponding to the sampled action is updated, while the other heads, built on top of the features that are getting updated, remain frozen. This is likely to contribute to the policy churn phenomenon identified by Schaul et al. (2022), highlighting that the greedy-policy changes for a significant proportion of the states in the replay buffer after a single batch update. To measure the impact of sharing features, we introduce the notion of *target churn*, which we define as the absolute value of the difference between the regression target before and after each batch update. We report the cumulative target churn of iS-DQN, reinitialized to zero after each target update, normalized by the target churn of TF-DQN in Figure 9 (middle). Conveniently, the target-based approach has a constant target churn of zero since the batch update does not influence the fully separated target network, and the normalization brings the target churn of the target-free approach to a constant value

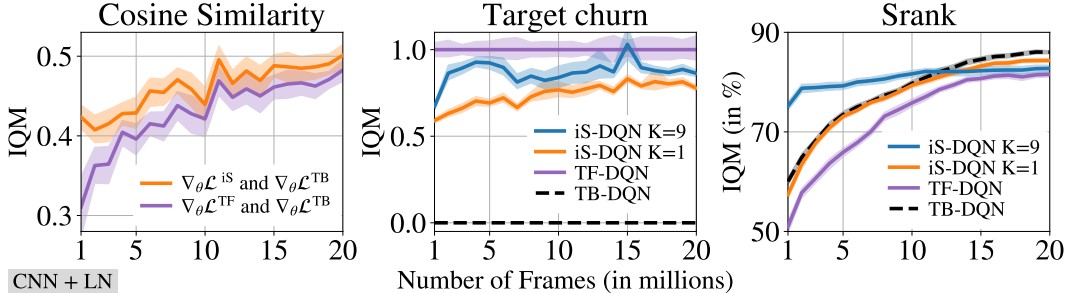

Figure 9: **Left**: The cosine similarity between the gradients w.r.t. the loss of iS-DQN and TB-DQN is larger than the cosine similarity between the gradients w.r.t. the loss of TF-DQN and TB-DQN. Therefore, iS-DQN brings the learning dynamics of the target-free approach closer to those of the target-based approach. **Middle**: The target churn is the difference between the regression targets computed before and after each batch update. The target predictions of iS-DQN are less influenced by batch updates than the ones computed from the target-free approach. **Right**: The effective rank (srank) of the features in the penultimate layer is higher for iS-QN, resulting in a higher expressivity.

of $1$. Remarkably, the target churn of iS-DQN $K = 1$ and $9$ lies in between $0$ and $1$, indicating that iS-QN's targets are more stable than the ones of the target-free approach. We note that the target churn for $K = 9$ is larger than $K = 1$, due to the influence of the additional terms in the loss.

Beyond improving the learning dynamics of TF-DQN, iS-DQN also provides a richer state representation. We measure the representation expressivity by reporting the effective rank (srank) of the features in the penultimate layer (Kumar et al., 2020a) in Figure 9 (right). Interestingly, the srank obtained by iS-DQN $K = 1$ is closer to the srank of TB-DQN than the srank of TF-DQN, which further demonstrates the benefit of using the last linear layer to construct the target. Notably, learning $K = 9$ Bellman iterations in parallel increases the representation capacity of the network by a large margin. This behavior is also visible in the offline setting, where iS-CQL reaches a similar srank as the target-based approach at the end of the training (see Figure 23, middle). This confirms the benefit of iS-QN to foster a richer representational capacity.

## 6 LIMITATION AND CONCLUSION

The proposed approach introduces the number of Bellman updates $K$ to learn in parallel as a new hyperparameter and there seems to be a different optimal value for each setting. However, we usually observe a stable increase in performance as $K$ grows until the benefit of the approach starts to diminish. Therefore, we recommend increasing $K$ until performance begins to decrease, as is commonly done with the learning rate. In Sections D.2, D.3, and D.4, we provide an extensive analysis of the dependency of iS-QN on its hyperparameters. In this work, we focus on reducing the memory footprint of the function approximators. Depending on the setting, other objects such as the replay buffer and the optimizer can occupy a large portion of the RAM. We remark that the proposed approach can be combined with other works addressing these issues (Vasan et al., 2024). Additionally, the proposed approach reduces the memory footprint during training but uses the same amount during inference, which is complementary to pruning methods that use more memory during training and less during inference (Graesser et al., 2022). As reported in Figure 11, iS-QN does not reduce the training time or the number of floating-point operations, except for the language processing task for which the temporal-difference error can be computed with a single pass through the network.

We introduced a simple yet efficient method for mitigating the performance drop that occurs when removing the target network in deep value-based reinforcement learning, while maintaining a low memory footprint. This is made possible by storing a copy of the last linear layer of the online network and using the features of the online network as input to this frozen linear head to construct the regression target. From there, more heads can be added to learn multiple Bellman iterations in parallel. We demonstrated that this new algorithm, iterated Shared $Q$-Networks, improves over the target-free approach and yields higher returns when the number of heads increases. We believe that combining iS-QN with mixed-precision training methods is a promising direction for future work to facilitate online learning in resource-constrained settings without sacrificing performance. In Section D.5, we provide a pilot study demonstrating positive results.

## ACKNOWLEDGMENTS

This research was supported by "Third Wave of AI", funded by the Excellence Program of the Hessian Ministry of Higher Education, Science, Research and Art, and by the grant "Einrichtung eines Labors des Deutschen Forschungszentrum für Künstliche Intelligenz (DFKI) an der Technischen Universität Darmstadt". The authors gratefully acknowledge the scientific support and HPC resources provided by the Erlangen National High Performance Computing Center (NHR@FAU) of the Friedrich-Alexander-Universität Erlangen-Nürnberg (FAU) under the NHR project b187cb. NHR funding is provided by federal and Bavarian state authorities. NHR@FAU hardware is partially funded by the German Research Foundation (DFG) – 440719683. This work was also partially supported by the German Federal Ministry of Research, Technology and Space (BMFTR) under the Robotics Institute Germany (RIG).

## REPRODUCIBILITY STATEMENT

Special care was taken to ensure this work is reproducible. **The code is available at `https://github.com/theovincent/iS-DQN`** and is shared in the supplementary material. It contains the list of dependencies and their exact version that was used to generate the results. To ease reproducibility, all hyperparameters are listed in Appendix E, and the individual training curves are shown in Appendix F.

## LARGE LANGUAGE MODEL USAGE

A large language model was helpful in polishing writing, improving reading flow, and identifying remaining typos.

## CARBON IMPACT

As recommended by Lannelongue & Inouye (2023), we used GreenAlgorithms (Lannelongue et al., 2021) and ML $CO_2$ Impact (Lacoste et al., 2019) to compute the carbon emission related to the production of the electricity used for the computations of our experiments. We only consider the energy used to generate the figures presented in this work and ignore the energy used for preliminary studies and for building the computing infrastructure. The estimations vary between 2.63 and 2.88 tonnes of $CO_2$ equivalent. As a reminder, the Intergovernmental Panel on Climate Change advocates a carbon budget of 2 tonnes of $CO_2$ equivalent per year per person.

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

## TABLE OF CONTENTS

## A   EXPERIMENT SETUP

**Atari setup** We build our codebase following Machado et al. (2018) standards and taking inspiration from Castro et al. (2018) codebase. Namely, we use the *game over* signal to terminate an episode instead of the life signal. The input given to the neural network is a concatenation of 4 frames in grayscale of dimension 84 by 84. To get a new frame, we sample 4 frames from the Gym environment (Brockman et al., 2016) configured with no frame-skip, and apply a max pooling operation on the 2 last grayscale frames. We use sticky actions to make the environment stochastic (with $p = 0.25$).

**Atari games selection** Our evaluations on the CNN architecture were performed on the 15 games recommended by Graesser et al. (2022). They were chosen for their diversity of Human-normalized score that DQN reaches after being trained on 200

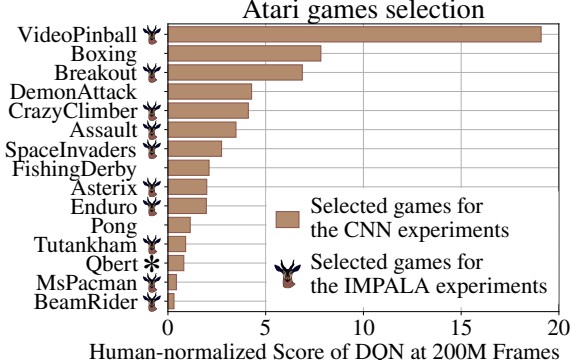

Figure 10: The Atari games selected for the experiments of this paper were chosen to cover a variety of normalized returns obtained by DQN after 200M frames. To lower the computational budget of the experiments with the IMPALA architecture, we reduced the set of games to 10 by removing 5 games, while maintaining diversity.

million frames, as shown in Figure 10. As the IMPALA architecture increases the training length, we removed 5 games, while maintaining diversity in the final scores to reduce the computational budget. For the offline experiment, we used the datasets provided by Gulcehre et al. (2020). As the game *Tutankham* is not available in the released dataset, we replaced it with *Qbert*, indicated with an asterisk in Figure 10. The codebase is adapted from the code released by Vincent et al. (2025a).

**DeepMind Control suite setup** Our codebase follows the implementation details of Lee et al. (2025). Before running the experiment presented in Section 5.3, we took special care that our codebase reproduces the evaluation performance shared by the authors. As a takeaway from this exercise, we note that the precision with which the state and reward are normalized matters, as using float32 leads to lower performance than using float64. We invite interested readers to examine our code for more details. We emphasize that the performances reported in this work correspond to those

collected during training, not the ones obtained during a separate evaluation phase, as they are closer to the initial motivation behind online learning (Machado et al., 2018).

**Wordle setup**     Our codebase is a fork of the repository shared by the authors (Snell et al., 2023), from which we implemented the target-free and the iterated Shared Features approaches. We refer to the original paper for extensive details about the setup. Every algorithm was given a budget of 600 000 gradient steps. This value was chosen by running TB-ILQL until the performance reported in Table 6 of Snell et al. (2023) was reached.

**Streaming RL setup**     We follow the implementation choices made by Elsayed et al. (2024). We detail here the differences with the classical Atari setup described earlier. Namely, the source of stochasticity is replaced from sticky actions to a random number of *NO OP* actions chosen uniformly between 0 and 30. The *life loss* signal is used for episode termination instead of the *game over* signal. The last hidden linear layer is composed of 256 neurons instead of 512. The neural network weights are initialized with a sparsity level of 90%. The optimizer's hyperparameters are $\kappa = 2$ and the default learning rate is 1. The trace coefficient is set to $\lambda = 0.8$.

**Computing the Area Under the Curve**     For each experiment, we report the normalized IQM AUC. For that, we first compute the undiscounted return obtained for each epoch, averaged over the episodes, as advocated by Machado et al. (2018). Then, we sum the human-normalized returns over the epochs and compute the IQM and 95% stratified bootstrap confidence intervals over the seeds and games. Finally, we divide the obtained values by the IQM of the target-based approach to facilitate the comparison. The human-normalized scores are computed from human and random scores that were reported in Schrittwieser et al. (2020). As discussed in Section 5, the normalized AUCs can also be interpreted as the average performance gap between the considered algorithm and the target-based approach. Indeed, dividing the two sums of performances across the training is equivalent to dividing the two averages of performances across the training because the normalizing factors cancel out.

## B  TRAINING TIME AND FLOATING-POINT OPERATIONS

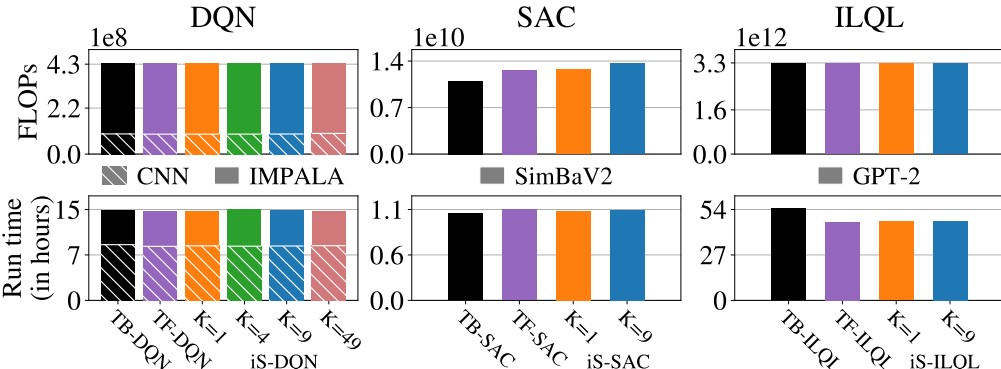

Figure 11: While TF-DQN and iS-DQN require fewer parameters, their training time is similar to TB-DQN since each algorithm uses a similar amount of computation, as indicated by the number of floating-point operations (FLOPs) per gradient steps. **Left**: All algorithms on the Atari benchmark require a similar amount of FLOPs and training time. **Middle**: As reported in Figure 25, the target-based approach does not benefit from BatchNorm for the DMC benchmark. This is why TB-SAC does not use BatchNorm and therefore has a lower amount of FLOPs compared to the other approaches. Importantly, the difference in training time between the algorithms is less visible across the algorithms. **Right**: Thanks to the way the embeddings are computed, the target-free approach and iS-ILQL can compute the TD error from a single pass through the neural network, which lowers the training time.

The presented approach is designed to reduce the memory footprint of target-based methods, while performing better than the target-free approach. In Figure 11 (bottom), we report the training time in hours required by all algorithms. On the top row, we report the number of floating-point operations (FLOPs) required by all algorithms to perform one gradient step. Computations were made on an NVIDIA GeForce RTX 4090 Ti with the game *Asterix* for the DQN experiments, and with the task *Dog-walk* for the SAC experiments. As expected, all algorithms require the same training time

and FLOPs because the same amount of computation is needed. Indeed, a forward pass through the network for estimating the value of the next state is necessary to compute the temporal-difference error. We note two exceptions. First, for experiments with SAC, the amount of FLOPs is reduced for the target-based approach, as it does not use BatchNorm. However, the difference in training time remains small. Second, for the experiments with ILQL, the training time for the target-free approach and iS-ILQL is smaller than that of the target-based approach since only one forward pass is required for computing the temporal difference instead of two forward passes. This results in the target-free approach and iS-ILQL having a small training time. While this reduction is not visible in the amount of FLOPs per gradient steps, we verified that the amount of FLOPs per loss computation (forward pass only) is indeed lower: TB-ILQL: $3.3 \times 10^{10}$ FLOPs, TF-ILQL: $2.0 \times 10^{10}$ FLOPs, iS-ILQL $K = 1$: $2.1 \times 10^{10}$ FLOPs, and iS-ILQL $K = 9$: $2.5 \times 10^{10}$ FLOPs.

We point out that when computing the algorithm's runtime, the environment steps are simulated, which does not reflect the real-world duration. For example, an episode of an Atari game can be executed significantly faster in simulation than in real life. This means that if real-world durations were taken into account, sample-efficient algorithms would achieve better performance earlier than less sample-efficient methods.

## C  ALGORITHMIC DETAILS

**Aggregating individual losses**     In Equation 1, we define the loss of iS-QN as the sum of losses over each Bellman iteration. Other ways of aggregating the losses are possible. Nonetheless, we decided to stick to the version proposed by Vincent et al. (2025b) and leave this investigation for future work. We provide a first alternative in Section 5.3 that provides a performance boost by discounting the following terms by a factor of $0.25$. While it is true that taking the sum of temporal differences increases the magnitude of the loss, it has a different impact on the updates than simply multiplying the learning rate by the number of terms in the loss. Indeed, the Adam optimizer (Kingma & Ba, 2015) first normalizes the gradient with a running statistic before applying the learning rate. Therefore, changing the aggregation mechanism has a greater impact on the direction of the update than on its magnitude. This is why we do not compare iS-QN against baselines instantiated with different learning rates.

**Sampling actions**     Following Vincent et al. (2025b), at each environment interaction, an action is sampled from a single head chosen uniformly as shown in Line 3 in Algorithm 1. The authors motivate this choice by arguing that it allows each $Q$-function to interact with the environment, thereby avoiding passive learning, identified by Ostrovski et al. (2021). This choice is further justified by an ablation study (see Figure 19 in Vincent et al. (2025b)) demonstrating a stronger performance against another sampling strategy consisting of sampling one head for each episode, as proposed in Osband et al. (2016).

In the experiment on continuous control (Section 5.3), the policy network is used to sample actions. To align with the choice of computing the discounted sum of temporal differences, the critic estimate in the policy loss is calculated as the average discounted prediction over the sequence of $Q$-predictions given by the heads. The experiment on the language task (Section 6) also uses a policy network to sample actions, but weighs each prediction with the predicted advantage from the critic. To align with the choice made for the experiment on continuous control, the average over the weights corresponding to each head is computed to obtain a single scalar value to weight each action probability.

# D  ADDITIONAL EXPERIMENTS

In this section, we train each agent for 40M frames rather than 100M to reduce the computational budget.

## D.1  COMPARISON WITH MELLOWMAX DQN

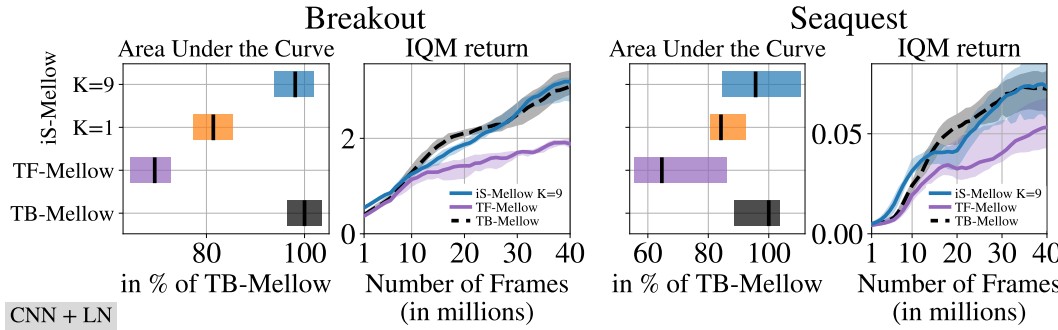

Figure 12: When using the MellowMax operator, the presented approach (iS-Mellow) effectively reduces the performance gap between the target-free approach (TF-Mellow) and the target-based approach (TB-Mellow).

We combine the MellowMax operator (Kim et al., 2019) with the presented idea. We select the games that were presented in the original paper (Kim et al., 2019), and use the same temperature coefficients ($\omega = 1\,000$ for *Breakout*, and $\omega = 30$ for *Seaquest*). Remarkably, in Figure 12, the performance gap between the target-free and target-based approaches is reduced by the presented approach.

## D.2  ABLATION STUDY ON THE TARGET UPDATE PERIOD

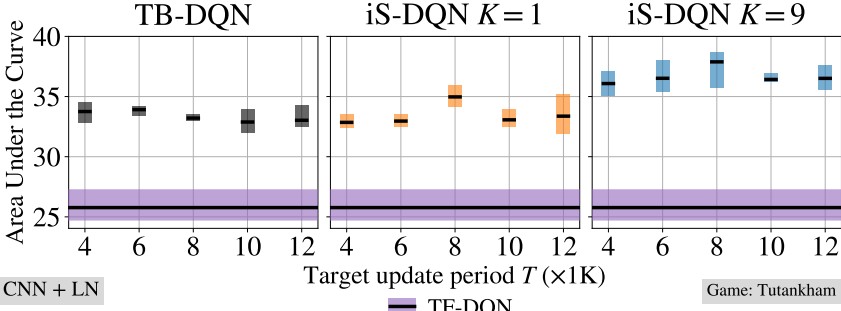

Figure 13: Ablation study on the target update period $T$ for the game *Tutankham*. iS-DQN's performance remains stable when varying the target update period. This shows that it is not sensitive to the target update period for this game.

We perform an ablation study on the frequency at which the heads of the iS-DQN agent are updated. This hyperparameter plays a role similar to the target update period $T$ in DQN, allowing both algorithms to propagate rewards, as explained in Section 4. In Figure 13, we report the performance of the target-based approach and the presented approach for 5 different values of $T$ on the game *Tutankham*. As a reminder, $T = 8000$ is the default value used in Section 5. We also report the performance of the target-free variant as a horizontal line since this algorithm does not use this hyperparameter. Interestingly, the performance of iS-DQN remains similar across different values of $T$, demonstrating its robustness to this hyperparameter in this game. Remarkably, iS-DQN $K = 9$ outperforms both the target-free and target-based variants across all target update periods.

### D.3 ABLATION STUDY ON THE NUMBER OF BELLMAN UPDATES

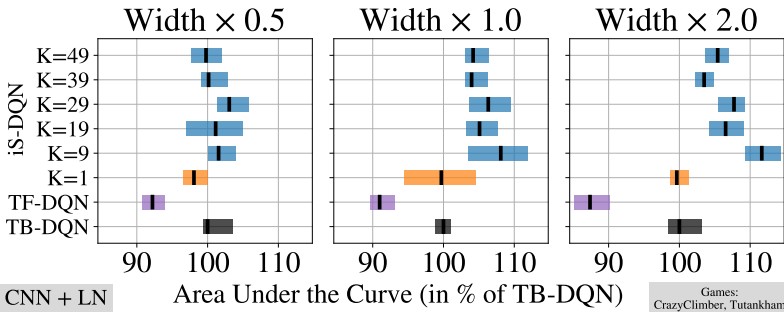

Figure 14: Ablation study on the number of Bellman updates learned in parallel $K$ and the size of the shared features. iS-DQN achieves similar performance for large values of $K$ and consistently outperforms the target-free approach.

We analyze the sensitivity of iS-DQN with respect to the number of Bellman updates $K$ learned in parallel. As discussed in Section 5.1, the richness of the shared features can affect the behavior of the method. To account for this aspect, we vary the size of the last linear layer from half the default value (256 neurons, "Width $\times$ 0.5") to twice the default (1024 neurons, "Width $\times$ 2.0"). To perform this study, we choose the games *CrazyClimber* and *Tutankham* as the performance of iS-DQN on those games for the default setting (512 neurons, "Width $\times$ 1.0") reflects the overall performance presented in Figure 3 on the 15 games. In Figure 14, we report the performance of iS-DQN for the three different settings with the number of heads varying from 2 ($K = 1$) to 50 ($K = 49$). As expected, the performance of iS-DQN is similar for large values of $K$ ($K \geq 9$). Importantly, every value of $K$ results in improved performance compared to the target-free variant, even if a slight decrease in performance occurs at the largest values of $K$. This is expected, as the potential of learning multiple Bellman updates in parallel can be better realized with the higher representational capacity.

### D.4 AUTOMATIC TUNING OF THE IMPORTANCE OF EACH BELLMAN UPDATE

In this section, we explore a way of tuning the importance of each Bellman update during training. For that, we propose weighting each term in the loss with a different coefficient. We note the coefficient $(\alpha_k)_{k=1}^{K}$. Intuitively, more weight should be given to a Bellman update that produces a gradient aligned with the gradients of the other Bellman updates. We will show that meta-learning those coefficients using the concept of meta-gradient reinforcement learning (Xu et al., 2018) provides a natural way to achieve this goal. Indeed, the update rule coming from the meta-gradient algorithm of a coefficient $\alpha_k$ linked to the $k^{\text{th}}$ Bellman update, depends on the dot product between the gradient of the loss of the $k^{\text{th}}$ Bellman update and the gradient of the sum of all Bellman updates:

$$\alpha_k \leftarrow \alpha_k + \lambda_\alpha \lambda_\theta \overbrace{\nabla_{\omega(\alpha)_k} \mathcal{L}_d^{\text{QN}}(\theta(\alpha)_k, \theta(\alpha)_{k-1})^T \nabla_{\omega_k} \mathcal{L}_d^{\text{QN}}(\theta_k, \theta_{k-1})}^{\substack{\text{How well aligned will the gradient of the loss} \\ \text{of the } k^{\text{th}} \text{ head be with its current value?}}}$$

$$+ \lambda_\alpha \lambda_\theta \underbrace{\nabla_{\omega(\alpha)} \left[ \sum_{i=1}^{K} \mathcal{L}_d^{\text{QN}}(\theta(\alpha)_i, \theta(\alpha)_{i-1}) \right]^T \nabla_\omega \mathcal{L}_d^{\text{QN}}(\theta_k, \theta_{k-1})}_{\substack{\text{How well aligned will the gradient of the overall loss w.r.t. the shared parameters} \\ \text{be with the value of the gradient of the } k^{\text{th}} \text{ head?}}}, \quad (2)$$

where $d = (s, a, r, s')$ is a sample, $\alpha$ is the vector of coefficients $(\alpha_k)_{k=1}^{K}$, $\theta(\alpha)$ is the parameters after one gradient step starting from $\theta$, $\omega$ is the shared parameters, $\omega_k$ is the parameters of the $k^{\text{th}}$ head such that $\theta_k = (\omega, \omega_k)$, $\lambda_\theta$ is the learning rate of the parameters, and $\lambda_\alpha$ is the meta learning rate.

In the following, we define the meta-optimization problem, analyse the empirical results, and finally derive Equation 2. To define the meta-optimization problem, we first stress the dependence of the learned parameters on the meta-parameters by noting them as a function of $\alpha$: $\theta^*(\alpha)$. Therefore, the inner loop of the meta-gradient optimization problem is defined as

$\theta^*(\alpha) = \min_\theta \sum_{k=1}^{K} \alpha_k \mathcal{L}_d^{\text{QN}}(\theta_k, \theta_{k-1})$. This leads to the following optimization problem:

$$\min_\alpha \sum_{k=1}^{K} \mathcal{L}_d^{\text{QN}}(\theta^*(\alpha)_k, \theta^*(\alpha)_{k-1}) \quad \text{s.t. } \theta^*(\alpha) = \min_\theta \sum_{k=1}^{K} \alpha_k \mathcal{L}_d^{\text{QN}}(\theta_k, \theta_{k-1})$$

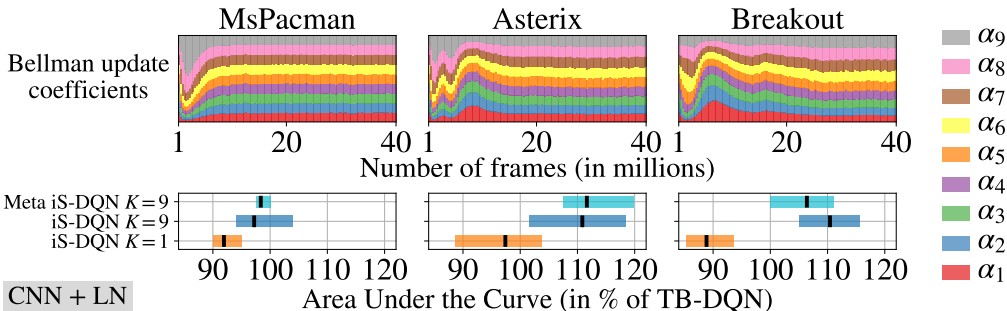

Figure 15: **Top**: Evolution of the meta-learned coefficients for each Bellman update during training for 3 Atari games. Remarkably, the coefficients converge to similar values, indicating that setting equal weights for each Bellman update is a good static choice for this setting. **Bottom**: Meta iS-DQN $K = 9$ performs on par with iS-DQN $K = 9$, which is coherent with the values of the learned coefficients.

We evaluate this approach for $K = 9$ on 3 Atari games, selected for their diversity in human-normalized scores (see Figure 10). We parameterize the meta-parameters $(\alpha_k)_{k=1}^{K}$ as logits $(z_k)_{k=1}^{K}$ and apply the softmax function such that the loss is a convex combination of the individual terms. In Figure 15 (top), we report the learned meta-parameters, which we use as a proxy for the importance of each Bellman update during training. Remarkably, after 10M frames, the coefficients converge to identical values, giving equal importance to each Bellman update. This indicates that setting equal weights for each Bellman update is a good static choice for this setting, as proposed in Equation 1. This explains why in Figure 15 (bottom), Meta iS-DQN $K = 9$ performs on par with iS-DQN $K = 9$, as iS-DQN $K = 9$ uses equal static weights during training. Importantly, Meta iS-DQN $K = 9$ performs better than iS-DQN $K = 1$.

To derive Equation 2, we assume that the meta-parameters only influence one update step, as in Xu et al. (2018). We also simplify the computations by choosing SGD as the optimizer instead of Adam, noting $\theta(\alpha)$ the parameters after one gradient step, starting from $\theta$. The meta-parameter update, for a sample $d = (s, a, r, s')$ is

$$\alpha_k \leftarrow \alpha_k - \lambda_\alpha \nabla_{\alpha_k} \sum_{i=1}^{K} \mathcal{L}_d^{\text{QN}}(\theta(\alpha)_i, \theta(\alpha)_{i-1}) = \alpha_k - \lambda_\alpha \sum_{i=1}^{K} \nabla_{\alpha_k} \mathcal{L}_d^{\text{QN}}(\theta(\alpha)_i, \theta(\alpha)_{i-1}), \quad (3)$$

$$\text{where } \nabla_{\alpha_k} \mathcal{L}_d^{\text{QN}}(\theta(\alpha)_i, \theta(\alpha)_{i-1}) = \nabla_{\theta(\alpha)_i} \mathcal{L}_d^{\text{QN}}(\theta(\alpha)_i, \theta(\alpha)_{i-1})^T \nabla_{\alpha_k} \theta(\alpha)_i$$
$$+ \nabla_{\theta(\alpha)_{i-1}} \mathcal{L}_d^{\text{QN}}(\theta(\alpha)_i, \theta(\alpha)_{i-1})^T \nabla_{\alpha_k} \theta(\alpha)_{i-1} \quad (4)$$

Equation 4 comes from the chain rule. We remark that for any $k$, $\nabla_{\theta_{k-1}} \mathcal{L}_d^{\text{QN}}(\theta_k, \theta_{k-1}) = 0$, since the target network is frozen. We split the parameters of the heads from the shared parameters as in Section 4: $\theta(\alpha)_i = (\omega(\alpha), \omega(\alpha)_i)$, where $\omega$ represents the shared parameters and $\omega_i$ the parameters of the $i^{\text{th}}$ head. We now focus on the gradient of the $i^{\text{th}}$ head parameters with respect to $\alpha_k$:

$$\nabla_{\alpha_k} \omega(\alpha)_i \overset{(a)}{=} \nabla_{\alpha_k} \left( \omega_i - \lambda_\theta \nabla_{\omega_i} \sum_{j=1}^{K} \alpha_j \mathcal{L}_d^{\text{QN}}(\theta_j, \theta_{j-1}) \right) \overset{(b)}{=} -\lambda_\theta \nabla_{\alpha_k} \left( \sum_{j=1}^{K} \alpha_j \nabla_{\omega_i} \mathcal{L}_d^{\text{QN}}(\theta_j, \theta_{j-1}) \right)$$

$$\overset{(c)}{=} -\lambda_\theta \nabla_{\alpha_k} \left( \alpha_i \nabla_{\omega_i} \mathcal{L}_d^{\text{QN}}(\theta_i, \theta_{i-1}) \right) \overset{(d)}{=} -\lambda_\theta \mathbb{1}_{k=i} \nabla_{\omega_i} \mathcal{L}_d^{\text{QN}}(\theta_i, \theta_{i-1}). \quad (5)$$

$(a)$ comes from the definition of $\omega(\alpha)_i$, $(b)$ uses the fact that $\omega_i$ and $\lambda_\theta$ do not depend on $\alpha_k$, $(c)$ uses the fact that the parameters of the $i^{\text{th}}$ head do not influence the parameters of the other heads, and $(d)$ uses the fact that $\nabla_{\alpha_k} \alpha_i = \mathbb{1}_{k=i}$. Similarly, we derive the gradient of the shared parameters $\omega(\alpha)$

with respect to $\alpha_k$:

$$
\nabla_{\alpha_k}\omega(\alpha) \overset{(e)}{=} \nabla_{\alpha_k}\left(\omega - \lambda_\theta \nabla_\omega \sum_{j=1}^{K}\alpha_j \mathcal{L}_d^{\mathrm{QN}}(\theta_j,\theta_{j-1})\right) \overset{(f)}{=} -\lambda_\theta \nabla_{\alpha_k}\left(\sum_{j=1}^{K}\alpha_j \nabla_\omega \mathcal{L}_d^{\mathrm{QN}}(\theta_j,\theta_{j-1})\right)
$$

$$
\overset{(g)}{=} -\lambda_\theta \left(\sum_{j=1}^{K}\nabla_{\alpha_k}\alpha_j \nabla_\omega \mathcal{L}_d^{\mathrm{QN}}(\theta_j,\theta_{j-1})\right) \overset{(h)}{=} -\lambda_\theta \left(\sum_{j=1}^{K}\mathbb{1}_{k=j}\nabla_\omega \mathcal{L}_d^{\mathrm{QN}}(\theta_j,\theta_{j-1})\right)
$$

$$
\overset{(i)}{=} -\lambda_\theta \nabla_\omega \mathcal{L}_d^{\mathrm{QN}}(\theta_k,\theta_{k-1}). \tag{6}
$$

$(e)$ comes from the definition of $\omega(\alpha)$, $(f)$ uses the fact that $\omega$ and $\lambda_\theta$ do not depend on $\alpha_k$, and the linearity of the sum, $(g)$ uses the linearity of the gradient operator, $(h)$ uses the fact that $\nabla_{\alpha_k}\alpha_j = \mathbb{1}_{k=j}$, and in $(i)$, we evaluate the indicator function. Plugging Equations 5 and 6 in Equation 4, we obtain:

$$
\nabla_{\alpha_k}\mathcal{L}_d^{\mathrm{QN}}(\theta(\alpha)_i,\theta(\alpha)_{i-1}) \overset{(j)}{=} \nabla_{\theta(\alpha)_i}\mathcal{L}_d^{\mathrm{QN}}(\theta(\alpha)_i,\theta(\alpha)_{i-1})^T \nabla_{\alpha_k}\theta(\alpha)_i
$$

$$
\overset{(k)}{=} \nabla_{\omega(\alpha)_i}\mathcal{L}_d^{\mathrm{QN}}(\theta(\alpha)_i,\theta(\alpha)_{i-1})^T \nabla_{\alpha_k}\omega(\alpha)_i + \nabla_{\omega(\alpha)}\mathcal{L}_d^{\mathrm{QN}}(\theta(\alpha)_i,\theta(\alpha)_{i-1})^T \nabla_{\alpha_k}\omega(\alpha)
$$

$$
\overset{(l)}{=} -\lambda_\theta \nabla_{\omega(\alpha)_i}\mathcal{L}_d^{\mathrm{QN}}(\theta(\alpha)_i,\theta(\alpha)_{i-1})^T \mathbb{1}_{k=i}\nabla_{\omega_i}\mathcal{L}_d^{\mathrm{QN}}(\theta_i,\theta_{i-1})
$$

$$
- \lambda_\theta \nabla_{\omega(\alpha)}\mathcal{L}_d^{\mathrm{QN}}(\theta(\alpha)_i,\theta(\alpha)_{i-1})^T \nabla_\omega \mathcal{L}_d^{\mathrm{QN}}(\theta_k,\theta_{k-1}) \tag{7}
$$

$(j)$ comes from Equation 4, $(k)$ uses the decomposition between the head parameters and the shared parameters, and $(l)$ comes from Equations 5 and 6. Finally, plugging Equation 7 in Equation 3 and using the linearity of the dot product, we obtain the result presented in Equation 2:

$$
\alpha_k \leftarrow \alpha_k + \lambda_\alpha \lambda_\theta \nabla_{\omega(\alpha)_k}\mathcal{L}_d^{\mathrm{QN}}(\theta(\alpha)_k,\theta(\alpha)_{k-1})^T \nabla_{\omega_k}\mathcal{L}_d^{\mathrm{QN}}(\theta_k,\theta_{k-1})
$$

$$
+ \lambda_\alpha \lambda_\theta \sum_{i=1}^{K}\nabla_{\omega(\alpha)}\mathcal{L}_d^{\mathrm{QN}}(\theta(\alpha)_i,\theta(\alpha)_{i-1})^T \nabla_\omega \mathcal{L}_d^{\mathrm{QN}}(\theta_k,\theta_{k-1}).
$$

$$
= \alpha_k + \lambda_\alpha \lambda_\theta \nabla_{\omega(\alpha)_k}\mathcal{L}_d^{\mathrm{QN}}(\theta(\alpha)_k,\theta(\alpha)_{k-1})^T \nabla_{\omega_k}\mathcal{L}_d^{\mathrm{QN}}(\theta_k,\theta_{k-1})
$$

$$
+ \lambda_\alpha \lambda_\theta \nabla_{\omega(\alpha)}\left[\sum_{i=1}^{K}\mathcal{L}_d^{\mathrm{QN}}(\theta(\alpha)_i,\theta(\alpha)_{i-1})\right]^T \nabla_\omega \mathcal{L}_d^{\mathrm{QN}}(\theta_k,\theta_{k-1}).
$$

### D.5    PILOT STUDY: MIXED PRECISION TRAINING OF iS-QN

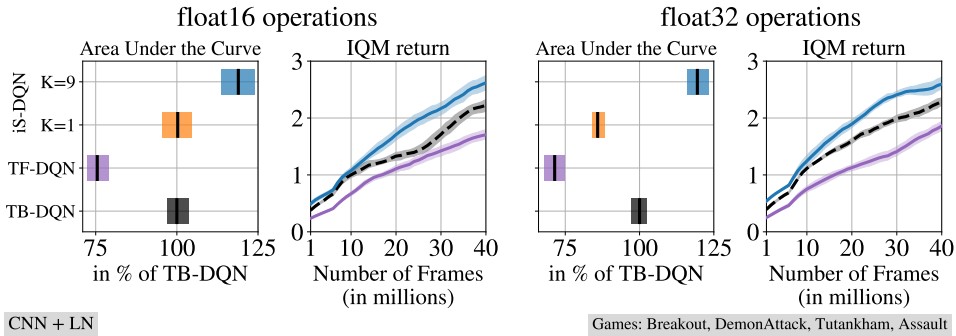

Figure 16: When reducing the precision of the operations performed during the forward and backward passes of the neural network from float32 to float16, iS-QN still bridges the performance gap between the target-free and target-based approaches.

To further reduce the resources required during training, the presented approach can be combined with other techniques that also aim at reducing resource requirements. We provide an initial study that combines the idea of performing operations with lower precision to further reduce the memory

footprint (Micikevicius et al., 2018) with the presented approach. Specifically, the computations during the forward and backward passes of the neural network are performed in float16b rather than float32. In Figure 16, we show that the performance gap between the target-free and target-based approaches remains when the operations are performed with lower precision. Remarkably, iS-DQN $K = 1$ closes the performance gap, and iS-DQN $K = 9$ further boosts the learning speed of the target-free approach, outperforming the target-based approach when the precision of the operations is reduced. Overall, we observe a slight decrease in performance across all algorithms, which is expected, as the precision of the operations during the forward and backward passes decreases.

## E  LIST OF HYPERPARAMETERS

Our codebase is written in Jax (Bradbury et al., 2018). The details of hyperparameters used for the experiments are provided in Table 1 (Atari), Table 2 (DMC Hard), and Table 3 (Wordle). In each experiment, the same hyperparameters as those provided in the original target-based approaches are used without further tuning. We note $\mathrm{Conv}^d_{a,b}C$ a $2D$ convolutional layer with $C$ filters of size $a \times b$ and stride $d$, and FC $E$ a fully connected layer with $E$ neurons. When added, LayerNorm is placed before each activation function, and BatchNorm is placed after the activation function. Additionally, when BatchNorm is used, the state-action and next state-next action pairs are first concatenated and then passed as a single batch to the network as suggested by the authors of CrossQ and CrossQ + WN (Bhatt et al., 2024; Palenicek et al., 2025).

Table 1: Summary of the shared hyperparameters used for the **Atari** experiments. The CNN architecture is described here. We used three stacked layers of size 32, 64, and 64 with a last linear layer of size 512 for the IMPALA architecture (Espeholt et al., 2018).

| Shared hyperparameters | |
|---|---|
| Discount factor $\gamma$ | 0.99 |
| Horizon $H$ | 27 000 |
| Full action space | No |
| Reward clipping | clip$(-1, 1)$ |
| Batch size | 32 |
| Torso architecture | $\mathrm{Conv}^4_{8,8}32$ $-\mathrm{Conv}^2_{4,4}64$ $-\mathrm{Conv}^1_{3,3}64$ |
| Head architecture | FC 512 $-$FC $n_{\mathcal{A}}$ [TB-QN, TF-QN] $-$FC $(K+1) \cdot n_{\mathcal{A}}$ [iS-QN] |
| Activations | ReLU |

| CQL hyperparameters | |
|---|---|
| Number of gradient steps per epoch | 62 500 |
| Target update period $T$ | 2 000 |
| Dataset size | 5 000 000 |
| Learning rate | $5 \times 10^{-5}$ |
| Adam $\epsilon$ | $3.125 \times 10^{-4}$ |
| CQL weight $\alpha$ | 0.1 |

| DQN hyperparameters | |
|---|---|
| Number of training steps per epoch | 250 000 |
| Target update period $T$ | 8 000 |
| Type of the replay buffer $\mathcal{D}$ | FIFO |
| Initial number of samples in $\mathcal{D}$ | 20 000 |
| Maximum number of samples in $\mathcal{D}$ | 1 000 000 |
| Gradient step period $G$ | 4 |
| Starting $\epsilon$ | 1 |
| Ending $\epsilon$ | 0.01 |
| $\epsilon$ linear decay duration | 250 000 |
| Batch size | 32 |
| Learning rate | $6.25 \times 10^{-5}$ |
| Adam $\epsilon$ | $1.5 \times 10^{-4}$ |

Table 2: Summary of the shared hyperparameters used for the **DMC Hard** experiments.

| Environment | |
|---|---|
| Discount factor $\gamma$ | 0.99 |
| Horizon $H$ | 1000 |
| Action repeat | 2 |
| **Experiments** | |
| Batch size | 256 |
| Policy architecture | SimbaV2 Actor |
| Critic Torso architecture | SimbaV2 Critic |
| Critic Head architecture | FC 512 $-$FC $n_{\text{atoms}}$ [TB-SAC, TF-SAC] $-$FC $(K+1) \cdot n_{\text{atoms}}$ [iS-SAC] |
| Activations | ReLU |
| BatchNorm | TF-SAC, iS-SAC |
| Number of training steps | 500 000 |
| Soft target update $\tau$ | $5 \times 10^{-3}$ |
| Initial number of samples in $\mathcal{D}$ | 5 000 |
| Maximum number of samples in $\mathcal{D}$ | 1 000 000 |
| Initial learning rate | $1 \times 10^{-4}$ |
| Final learning rate | $1 \times 10^{-5}$ |
| Optimizer | Adam |
| **SimbaV2 hyperparameters** | |
| Double Q | No |
| Distributional critic bins $n_{\text{atoms}}$ | 101 |

Table 3: Summary of the shared hyperparameters used for the **Wordle** experiments.

| Environment | |
|---|---|
| Dataset | Wordle Twitter dataset |
| Discount factor $\gamma$ | 0.99 |
| Number of tokens | 35 (alphabet + colors) |
| Rewards | $-1$ for incorrect guess, 0 for correct guess |
| **Experiments** | |
| Batch size | 1024 |
| Policy architecture | GPT-2 small (Dropout $p = 0.1$) |
| Torso architecture $Q, V$ | GPT-2 small (Dropout $p = 0.1$) |
| Head architecture $Q$ | FC 1536 $-$FC $n_{\mathcal{A}}$ [TB-ILQL, TF-ILQL] $-$FC $(K+1) \cdot n_{\mathcal{A}}$ [iS-ILQL] |
| Head architecture $V$ | FC 1536 $-$FC 1 [TB-ILQL, TF-ILQL] $-$FC $(K+1) \cdot 1$ [iS-ILQL] |
| Activations | ReLU |
| Number of gradient steps | 600 000 |
| Soft target update $\tau$ | $5 \times 10^{-3}$ |
| Learning rate | $1 \times 10^{-5}$ |
| Optimizer | Adam |
| **ILQL hyperparameters** | |
| Inverse temperature $\beta$ | 4.0 |
| CQL weight $\alpha$ | $1 \times 10^{-4}$ |

## F INDIVIDUAL LEARNING CURVES

### F.1 DEEP $Q$-NETWORK WITH CNN AND LAYERNORM

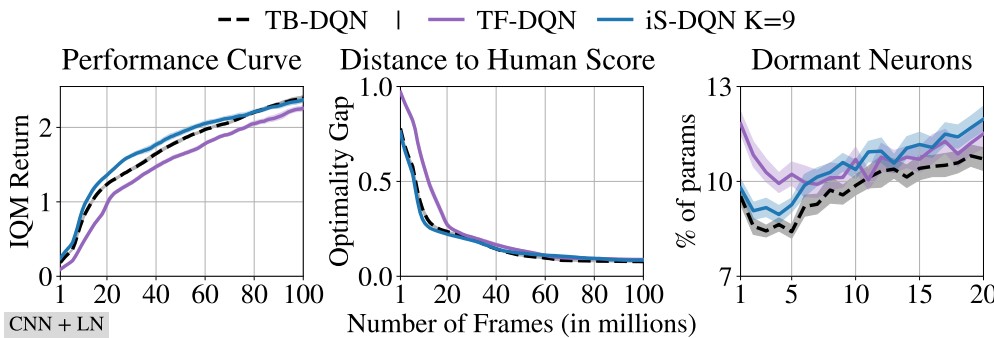

Figure 17: Reducing the performance gap in online RL on 15 **Atari** games with the CNN architecture and LayerNorm. **Left**: iS-DQN $K = 9$ not only reduces the performance gap but outperforms the target-based approach. **Middle**: iS-DQN annuls the performance gap for the games where the score is below the human level. **Right**: iS-DQN exhibits a lower amount of dormant neurons at the beginning of the training compared to the target-free approach.

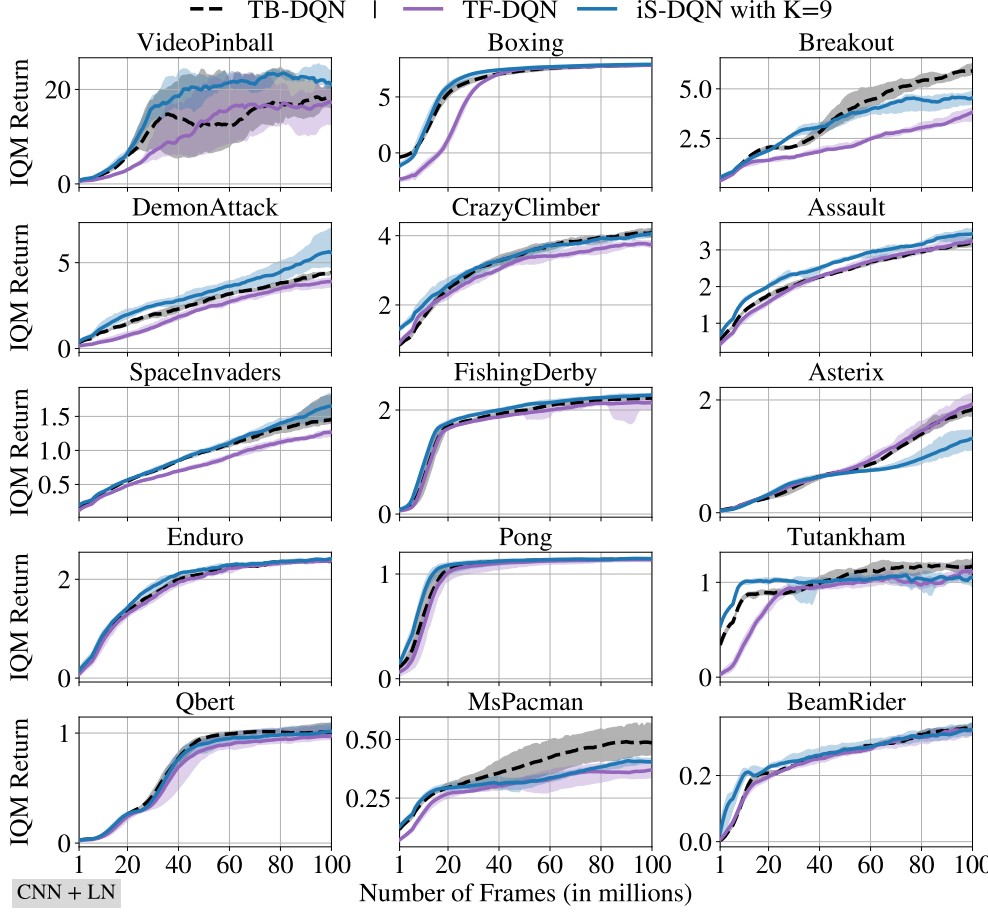

Figure 18: Per game training curves of iS-DQN, TF-DQN, and TB-DQN with the CNN architecture and LayerNorm. Except on *Asterix*, iS-DQN outperforms or is on par with the target-free approach (TF-DQN).

## F.2 DEEP $Q$-NETWORK WITH IMPALA AND LAYERNORM

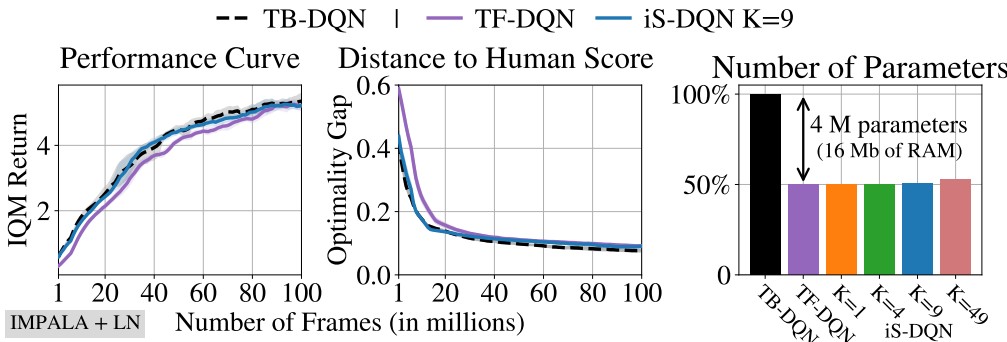

Figure 19: Reducing the performance gap in online RL on 10 **Atari** games with the IMPALA architecture and LayerNorm. **Left**: iS-DQN $K = 9$ is outperforms the target-free approach. **Middle**: iS-DQN annuls the performance gap for the games where the score is below the human level. **Right**: iS-DQN requires significantly fewer parameters than the target-based approach while reaching similar performance.

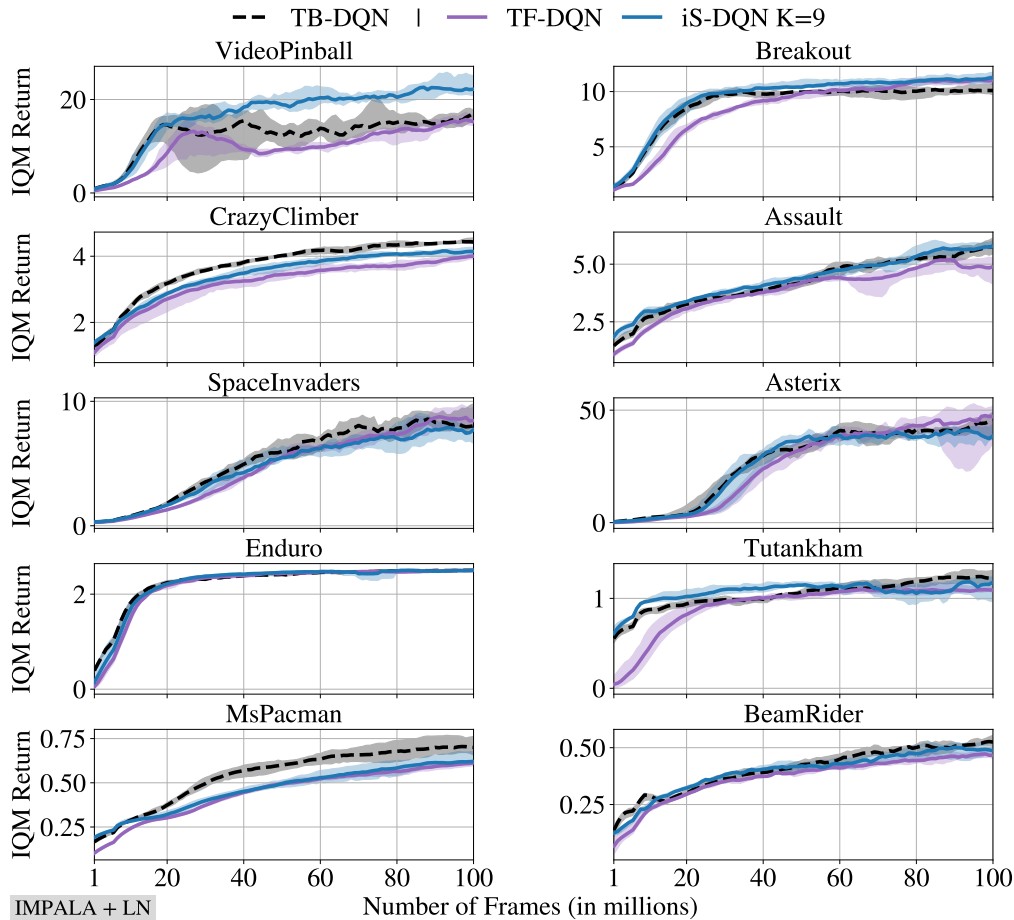

Figure 20: Per game training curves of iS-DQN, TF-DQN, and TB-DQN with the IMPALA architecture and LayerNorm. Our approach outperforms or is on par with the target-free approach (TF-DQN) on all games.

### F.3 DEEP $Q$-NETWORK WITH CNN

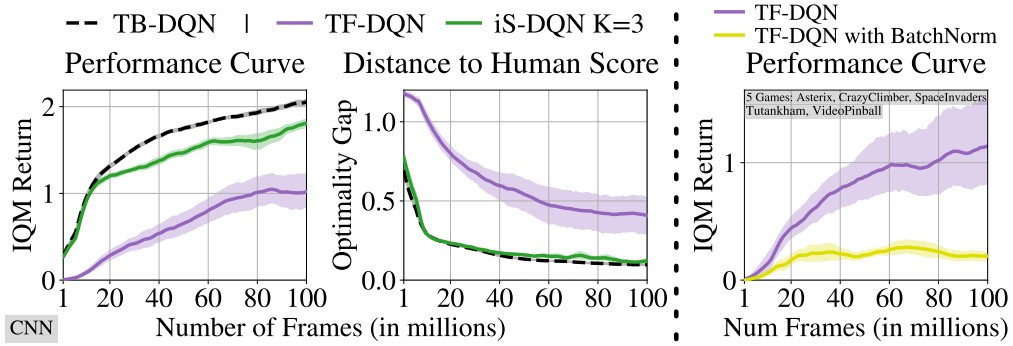

Figure 21: Reducing the performance gap in online RL on 15 **Atari** games with the CNN architecture. **Left**: iS-DQN $K = 3$ significantly reduces the performance gap between the target-free and target-based approaches. **Middle**: iS-DQN annuls the performance gap for the games where the score is below the human level. **Right**: Including BatchNorm in the architecture damages the performance on the 5 considered games of the target-based approach. This is why BatchNorm is not included for the experiments with TB-DQN.

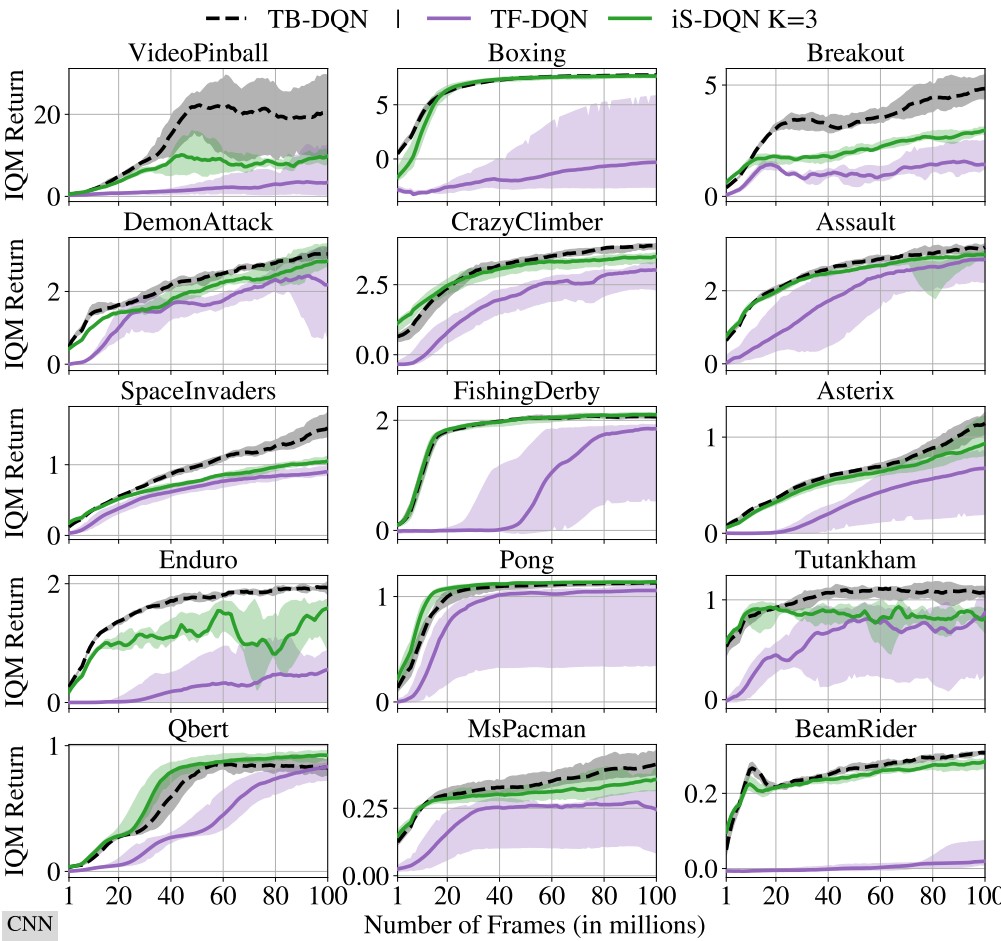

Figure 22: Per game training curves of iS-DQN, TF-DQN, and TB-DQN with the CNN architecture. Remarkably, iS-DQN outperforms the target-free approach (TF-DQN) on all games.

## F.4 CONSERVATIVE $Q$-LEARNING WITH IMPALA AND LAYERNORM

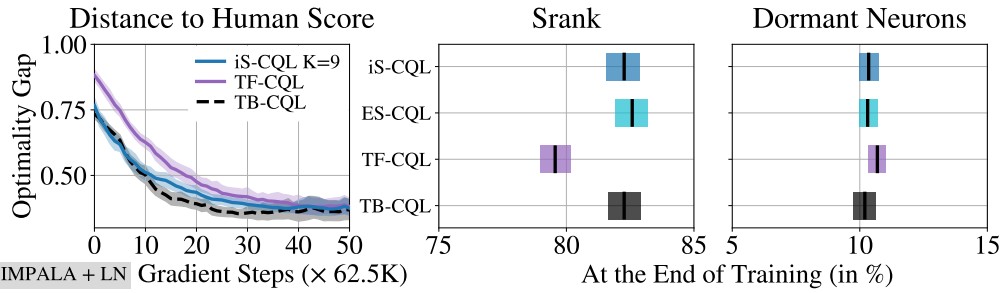

Figure 23: Reducing the performance gap in offline RL on 10 **Atari** games with the IMPALA architecture and LayerNorm. **Left**: iS-CQL significantly reduces the performance gap for the gameS where the score is below the human level. **Middle**: At the end of the training, iS-CQL and ES-CQL lead to a higher srank than the target-free approach, which indicates a higher representation capability. **Right**: All methods converge to a low amount of dormant neurons at the end of the training.

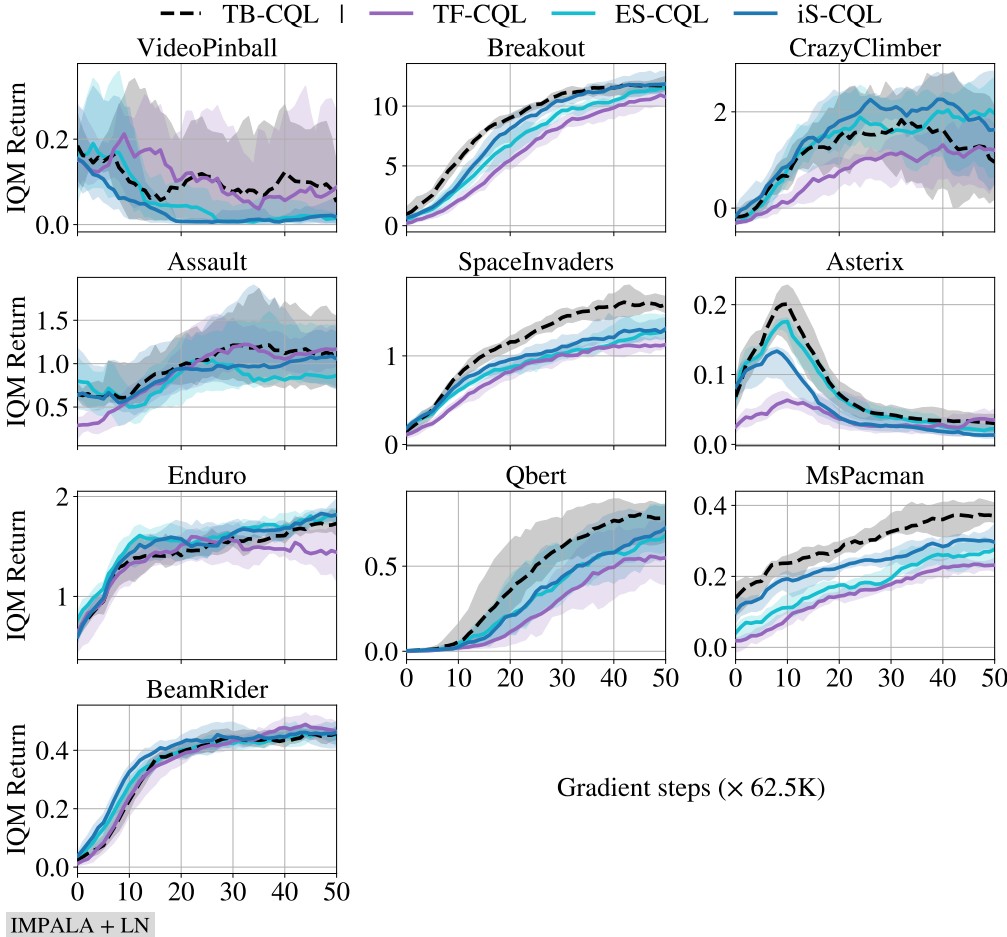

Figure 24: Per game training curves of iS-CQL, TF-CQL, and TB-CQL with the IMPALA architecture and LayerNorm. Except on *VideoPinball*, iS-CQL outperforms or is on par with the target-free approach (TF-CQL).

### F.5 SOFT ACTOR-CRITIC WITH SIMBAV2 AND BATCHNORM

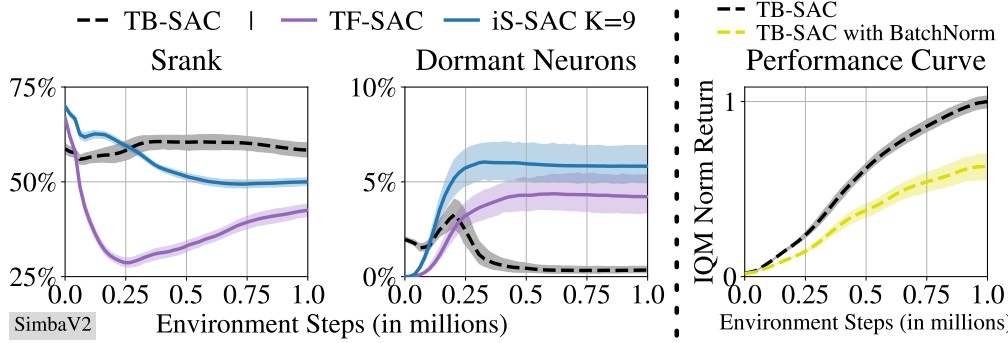

Figure 25: Reducing the performance gap in online RL on the 7 **DMC Hard** tasks with the SimbaV2 architecture and BatchNorm. **Left**: As opposed to iS-SAC, the target-free approach suffers from a low srank, which indicates a lower representation capability. **Middle**: The percentage of dormant neurons remains low during training for all methods, not exceeding 7%. **Right**: The target-based approach does not benefit from BatchNorm. This is why it is not included in the experiments with TB-SAC. Importantly, all algorithms use $\ell_2$-norm as described in Lee et al. (2025).

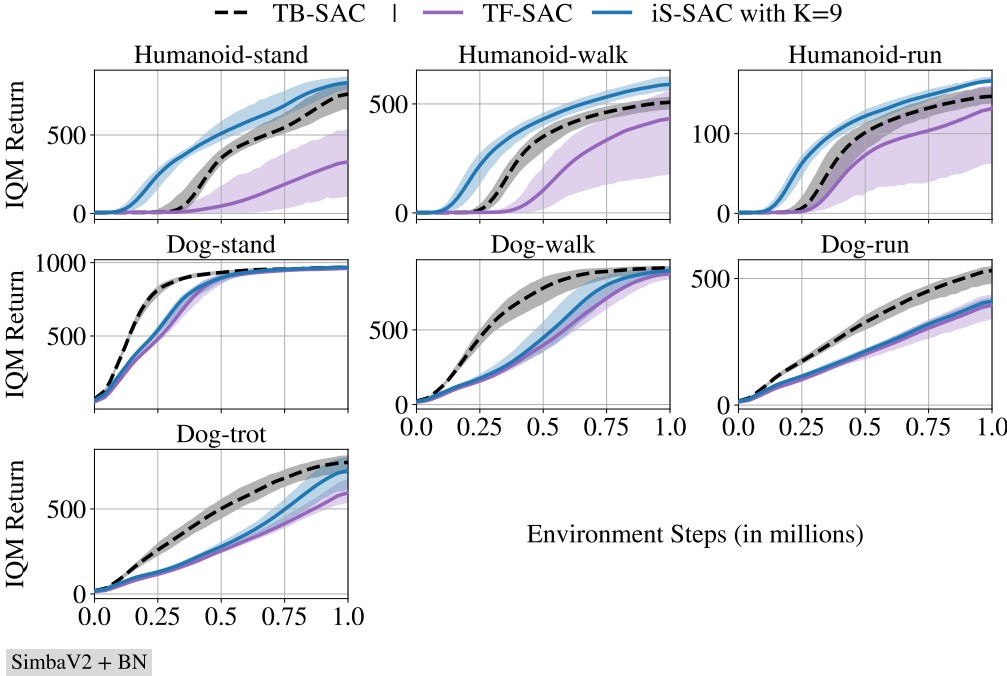

Figure 26: Per task training curves of iS-SAC, TF-SAC, and TB-SAC with the SimbaV2 architecture and BatchNorm. iS-SAC consistently performs better than or on par with the target-free approach. Interestingly, iS-SAC even outperforms the target-based approach on the humanoid tasks.

