# OpenReview forum: "Bridging the performance-gap between target-free and target-based reinforcement learning"
_ICLR.cc/2026/Conference — ICLR 2026 Poster_

### Official Review · Reviewer_Bpoz · 2025-10-15

**Soundness:** 2
**Presentation:** 2
**Contribution:** 2
**Rating:** 4
**Confidence:** 3

**Summary:**

This paper presents iterated Shared Q-Learning, a deep reinforcement learning method designed to bridge the gap between target-free and target-based approaches. The proposed technique stores only the last linear layer of an old Q-network as a frozen target, while sharing the remaining parameters with the current online network. Coupling this shared features innovation with iterated Q-learning reduces the memory footprint compared to classic target-network approaches without sacrificing stability. Extensive experiments (Atari, DMC continuous control, language model RL) show that iS-QN recovers much of the performance lost by omitting target networks, sometimes even outperforming classic methods in both online and offline settings.

**Strengths:**

1. The paper is well-motivated, as reusing all but the final linear layer for target calculation cuts memory usage, making deep RL feasible even on resource-constrained hardware or for large architectures.

2. The framework is tested on diverse discrete and continuous control tasks, as well as language-based RL, complemented with analyses on training dynamics, expressivity of representations, and the stability of regression targets.

**Weaknesses:**

1. The number of parallel Bellman updates (heads, K) needs tuning, and optimal values may vary with domain and architecture.

2. While memory use is reduced, floating-point operations and wall-clock training time are not improved compared to baseline methods.

**Questions:**

1. Is there a way to adaptively select or anneal K during training?

2. Can you answer W2? Adding studies or discussions on continual learning/resource-constrained real-world deployments might be a good idea.

---

> ### Author Response · Authors · 2025-11-26
>
> We thank the Reviewer for the time spent reviewing the presented work and for the insightful comments.
>
> ## Weaknesses
>
> > I. The number of parallel Bellman updates (heads, K) needs tuning, and optimal values may vary with domain and architecture.
>
> In all settings, we observe that the performance increases when considering more heads, until a peak in performance is reached. Therefore, in Section $6$ of the revised submission, we provide some tuning guidance, advising to increase this hyperparameter until performance starts to decrease, as is commonly done with the learning rate. Furthermore, in Section $D.3$, we provide an ablation study on the number of parallel Bellman updates ($K$) and the size of the last linear layer (the one that is shared across the heads). We observe that iS-DQN outperforms the target-free variant for all considered values of $K$, across all shared feature sizes. Interestingly, better performances are obtained when the number of shared features increases. This is expected, as the potential of learning multiple Bellman updates in parallel can be better realized with the higher representational capacity.
>
> > II. While memory use is reduced, floating-point operations and wall-clock training time are not improved compared to baseline methods.
>
> In this work, we focus on learning speed in terms of the number of environment interactions, as this is a major bottleneck in many real-world scenarios. We point out that when computing the algorithm's runtime, the environment steps are simulated, which does not reflect the real-world duration. For example, an episode of an Atari game can be executed significantly faster in simulation than in real life. This means that if real-world durations were taken into account, sample-efficient algorithms would achieve better performance significantly earlier than less sample-efficient methods. We clarify this point in Section $B$ of the revised submission.
>
> ## Questions
> > 1. Is there a way to adaptively select or anneal K during training?
>
> We thank the Reviewer for the suggestion. It is possible to adaptively tune the importance of each Bellman update in the loss. Indeed, we added a study explaining how to do it in Section $D.4$ of the revised submission.
>
> Intuitively, more weight should be given to a Bellman update that produces a gradient aligned with the gradients of the other Bellman updates. We show that meta-learning those coefficients using the concept of meta-gradient reinforcement learning provides a natural way to achieve this goal. We evaluate this method on $3$ Atari games. This allows us to analyse the evolution of the coefficients during training. Interestingly, the coefficients converge rapidly to a static value, giving equal weight to each Bellman update. This indicates that setting equal weights for each Bellman update is a good static choice for this setting. Importantly, the adaptive approach, Meta iS-DQN $K = 9$, performs better than iS-DQN $K = 1$.
>
> Overall, as meta-gradient reinforcement learning usually requires a larger compute budget, we believe this adaptive approach should primarily be used to understand the method's behavior by analyzing the evolution of the coefficients during training.
>
> > 2. [...] Adding studies or discussions on continual learning/resource-constrained real-world deployments might be a good idea.
>
> In Section $5.5$ of the revised submission, we added a new experiment in the streaming reinforcement learning setting. This setting falls under the incremental learning setting with resource constraints, as the agent learns from a stream of data without a replay buffer or batch updates. Those constraints are imposed on the learning algorithm to develop methods more aligned with real-world scenarios. We show that the presented method can be combined with an existing method [1] to further improve the learning speed of the target-free approach. This demonstrates that the presented approach is also relevant in the streaming scenario.
>
> [1] Elsayed, Mohamed, Gautham Vasan, and A. Rupam Mahmood. "Streaming deep reinforcement learning finally works." arXiv preprint arXiv:2410.14606 (2024).

---

### Official Review · Reviewer_959t · 2025-10-28

**Soundness:** 2
**Presentation:** 2
**Contribution:** 2
**Rating:** 6
**Confidence:** 3

**Summary:**

This paper proposes Iterated Shared Q-Network (iS-QN), a simple method that stores only the final linear layer of the target network while sharing all other layers with the online network. Combined with iterated Q-learning, iS-QN learns multiple Bellman updates in parallel, reducing memory use and improving sample efficiency. Experiments across Atari, DMC, and Wordle tasks show that iS-QN bridges or surpasses the performance gap between target-free and target-based RL with minimal additional cost.

**Strengths:**

The notion of retaining only the last linear layer of the target network (shared features) is conceptually simple but impactful. It effectively merges the strengths of both paradigms without introducing significant architectural or computational overhead.
Moreover, the evaluation spans multiple benchmarks (Atari, DMC, Wordle), architectures (CNN, IMPALA, SimbaV2, GPT-2), and settings (online/offline/continuous).
The breadth of validation strongly supports the claim of generality and practicality.
The paper achieves significant empirical improvements over many benchmark examples.
Detailed hyperparameters, FLOP analyses (Fig. 10), and appendices ensure reproducibility.
Code release and dependency listings are also promised upon acceptance

**Weaknesses:**

Despite solid empirical results, the paper provides no formal convergence or stability analysis. For instance, how the partial sharing of target parameters affects the contraction properties of the Bellman operator remains unaddressed.

The optimal number of parallel Bellman updates (K) varies across domains (e.g., K = 9 works for Atari, K = 1 for SAC). The paper admits this (Sec. 6) but does not analyze the scaling law or provide tuning guidance.
While works like Kim et al. (2019), Bhatt et al. (2024), and Gallici et al. (2025) are cited, the paper lacks direct quantitative comparison under identical settings.
The FLOP and runtime discussion (Fig. 10) shows that iS-QN doesn’t actually reduce compute per iteration. Clarifying whether sample efficiency translates into wall-clock speed-up would strengthen the practical argument.
The exposition occasionally interchanges terms like iS-QN, iS-DQN, iS-CQL, etc., without immediate redefinition. A concise schematic or unified notation table would improve readability.

**Questions:**

1. What is the empirical or theoretical trade-off between K and variance/stability?
2. How does feature sharing interact with normalization layers (LayerNorm vs BatchNorm) beyond empirical observation?
3. Can the authors provide ablation on the frequency T of head updates, similar to target-network refresh rate in DQN?
4. Future work: combining iS-QN with network pruning or quantization could yield compelling results for edge RL; a brief pilot experiment would add value.

---

> ### Author Response · Authors · 2025-11-26
> **Answers to Weaknesses**
>
> We thank the Reviewer for the extensive feedback and numerous suggestions. We integrated the feedback in the revised version of the submission.
>
> ## Weaknesses
>
> > I. Despite solid empirical results, the paper provides no formal convergence or stability analysis. For instance, how the partial sharing of target parameters affects the contraction properties of the Bellman operator remains unaddressed.
>
> While this work is based on a theoretical understanding of temporal-difference learning, our contribution is deeply involved with non-linear function approximation, which significantly increases the difficulty of deriving meaningful theoretical results.
>
> Nonetheless, we refer to Section $5.6$, which provides an analysis to better understand the behavior of our approach. First, in Figure $9$ (left), we show that the gradients obtained with iS-DQN $K=1$ point in a similar direction to the gradient obtained with the target-based version. Then, in Figure $9$ (middle), we show that the change in the target prediction remains smaller for iS-DQN than for the target-free version, indicating more stable behavior. Finally, in Figure $9$ (right), we show that adding additional heads increases the richness of the state representation.
>
> > II. The optimal number of parallel Bellman updates (K) varies across domains (e.g., K = 9 works for Atari, K = 1 for SAC). The paper admits this (Sec. 6) but does not analyze the scaling law or provide tuning guidance.
>
> We kindly ask the Reviewer to refer to our answer for Question $1$, which addresses this point.
>
> > III. While works like Kim et al. (2019), Bhatt et al. (2024), and Gallici et al. (2025) are cited, the paper lacks direct quantitative comparison under identical settings.
>
> We thank the Reviewer for the suggestion. Kim et al. ($2019$), Bhatt et al. ($2024$), and Gallici et al. ($2025$) introduced techniques that enable learning without target networks.
>
> Kim et al. ($2019$) introduced the MellowMax operator to combat instabilities. In Section $D.1$ of the revised submission, we demonstrate that the presented approach bridges the performance gap between the target-free and target-based approaches when the MellowMax operator is used instead of the maximum operator.
>
> Bhatt et al. ($2024$) showed that BatchNorm is beneficial for learning without a target network in continuous action settings. Therefore, we incorporated it when evaluating the target-free approach in Section $5.3$, as mentioned in the last paragraph of Section $4$.
>
> Gallici et al. ($2025$) showed that LayerNorm is beneficial for learning without a target network in discrete action settings. Therefore, we incorporated it when evaluating the target-free approach in Section $5$, as mentioned in the last paragraph of Section $4$.
>
>
> > IV. The FLOP and runtime discussion (Fig. 10) shows that iS-QN doesn’t actually reduce compute per iteration. Clarifying whether sample efficiency translates into wall-clock speed-up would strengthen the practical argument.
>
> In this work, we focus on learning speed in terms of the number of environment interactions, as this is a major bottleneck in many real-world scenarios. We point out that when computing the algorithm's runtime, the environment steps are simulated, which does not reflect the real-world duration. For example, an episode of an Atari game can be executed significantly faster in simulation than in real life. This means that if real-world durations were taken into account, sample-efficient algorithms would achieve better performance significantly earlier than less sample-efficient methods. We clarify this point in Section $B$ of the revised submission.
>
> > V. The exposition occasionally interchanges terms like iS-QN, iS-DQN, iS-CQL, etc., without immediate redefinition. A concise schematic or unified notation table would improve readability.
>
> We thank the Reviewer for the suggestion. We clarified this aspect in the last paragraph of Section $4$.

---

> > ### Author Response · Authors · 2025-11-26
> > **Answers to Questions**
> >
> > ## Questions
> >
> > > 1. What is the empirical or theoretical trade-off between K and variance/stability?
> >
> > In all settings, we observe that the performance increases when considering more heads, until a peak in performance is reached. Therefore, in Section $6$ of the revised submission, we provide some tuning guidance, advising to increase this hyperparameter until performance starts to decrease, as is commonly done with the learning rate. Furthermore, in Section $D.3$, we provide an ablation study on the number of parallel Bellman updates ($K$) and the size of the last linear layer (the one that is shared across the heads). We observe that iS-DQN outperforms the target-free variant for all considered values of $K$, across all shared feature sizes. Interestingly, better performances are obtained when the number of shared features increases. This is expected, as the potential of learning multiple Bellman updates in parallel can be better realized with the higher representational capacity.
> >
> > > 2. How does feature sharing interact with normalization layers (LayerNorm vs BatchNorm) beyond empirical observation?
> >
> > The normalization layers in the architecture of the neural network constrain the values of the shared features, thus ensuring that a few heads do not dominate the gradient w.r.t. shared parameters, which could result in overfitting. We stress that layer normalization was added to strengthen the target-free approach by incorporating insights from related work. We believe that investigating the interaction between the normalization layers and feature sharing goes beyond the scope of this work.
> >
> > > 3. Can the authors provide ablation on the frequency T of head updates, similar to target-network refresh rate in DQN?
> >
> > We thank the Reviewer for the suggestion. We added an ablation study on the target update period in Section $D.2$. iS-DQN with $K=1$ and $K=9$ perform similarly across different target update periods, consistently outperforming the target-free approach.
> >
> > > 4. Future work: combining iS-QN with network pruning or quantization could yield compelling results for edge RL; a brief pilot experiment would add value.
> >
> > We thank the Reviewer for the suggestion. In Section $D.5$ of the revised submission, we added a pilot study on $4$ Atari games, showing that the presented method keeps its benefits when using a lower precision (float$16$b instead of float$32$) during the forward pass and backward pass of the neural network to reduce memory footprint further, as introduced in [1]. We adapted the last sentence of the conclusion accordingly.
> >
> > [1] Micikevicius et al. Mixed precision training. ICLR, 2018.

---

### Official Review · Reviewer_Hr47 · 2025-10-31

**Soundness:** 3
**Presentation:** 4
**Contribution:** 3
**Rating:** 4
**Confidence:** 2

**Summary:**

The paper proposes an alternative to target networks, namely keeping only a frozen copy of the last linear layer as the “target,” while sharing all upstream features with the online network. Building on this, the authors use iterated Shared Q‑Networks (iS‑QN), a method that uses multiple linear heads that learn consecutive Bellman updates in parallel. Across Atari (online/offline), DMC Hard (SAC), and Wordle (ILQL), this closes or even beats target‑based baselines while using far less memory. The paper also offer some empirical explainations why the method stabilizes learning.

**Strengths:**

The main strength of this work lies in the problem they are solving, which is removing the target network from deep value based RL methods. Their idea can be integrated into DQN code with minimal changes while preserving the memory usage of a target free method and enjoying the stability of a target based method. The experiments appear extensive and well executed.

**Weaknesses:**

The main weakness lie in this paper's similarity to that of Elsayed 2024 [1] and the follow up by Vasan 2025, which also claims to solve a similar problem, this work would benefit greatly from a more direct comparison with these works. By properly placing your contributions in the context of these works, it would greatly enhance and appropriately detail the contributions of your work.

While somewhat orthogonal to this work, it would be nice if the authors could give some theoretical explains (even on simple mdps/toy problems) as to why their method enjoys a best of both worlds (target free memory, target based stability) thought this might be hard with deep networks.

[1] Elsayed, Mohamed, Gautham Vasan, and A. Rupam Mahmood. "Streaming deep reinforcement learning finally works." arXiv preprint arXiv:2410.14606 (2024).

**Questions:**

1. How does your work compare with the work of Elsayed 2024 [1] and Vasan 2025, which as you mention in your work, remove the need for the target network? When/why should one be preferred over the other? Why did you not compare your method with theirs?

Given this is my main issue with this work, I would consider raising my score if this is adequately addressed.

---

> ### Author Response · Authors · 2025-11-26
>
> We thank the Reviewer for the suggestions and the time spent reviewing the presented work.
>
> ## Weaknesses
>
> > I. The main weakness lie in this paper's similarity to that of Elsayed 2024 [1] and the follow up by Vasan 2025, which also claims to solve a similar problem [...]
>
> We kindly ask the Review to refer to our answer to Question $1$, which addresses this point.
>
> > II. While somewhat orthogonal to this work, it would be nice if the authors could give some theoretical explains [...] as to why their method enjoys a best of both worlds [...] thought this might be hard with deep networks.
>
> While our work is based on a theoretical understanding of temporal difference learning, we do not provide a theoretical analysis of the proposed approach because our contribution is linked with the architecture of the function approximator. As acknowledged in the Review, the use of deep networks significantly increases the difficulty of deriving meaningful results.
>
> Nonetheless, we refer to Section $5.6$, which provides an analysis to better understand the behavior of our approach. First, in Figure $9$ (left), we show that the gradients obtained with iS-DQN $K=1$ point in a similar direction to the gradient obtained with the target-based version. Then, in Figure $9$ (middle), we show that the change in the target prediction remains smaller for iS-DQN than for the target-free version, indicating more stable behavior. Finally, in Figure $9$ (right), we show that adding additional heads increases the richness of the state representation.
>
> Overall, the foundation of our approach relies on the idea of sharing the last features and using multiple heads to represent the different quantities that are typically approximated by independent networks. As explained in Section $5.6$, this idea is also used in DQN to avoid learning one neural network for each action-value prediction. This choice is made to reduce the number of parameters while still reaching high performance. At each gradient step, one head is updated while the others remain frozen. This work aims to apply the same logic to the target and online networks. To further boost the learning speed, we combine this idea with iterated $Q$-learning, which leads to a parameter-efficient algorithm that is competitive with the target-based approach.
>
> ## Questions
>
> > 1. How does your work compare with the work of Elsayed 2024 [1] and Vasan 2025, which as you mention in your work, remove the need for the target network? When/why should one be preferred over the other? Why did you not compare your method with theirs?
>
> We thank the Reviewer for the suggestion. We stress that the method presented in [1] is designed for the streaming reinforcement learning setting. This means that the authors are trying to solve a different problem: not only is the target network removed, but, to further reduce the required resources, the batch updates and the replay buffer are also not accessible to the learning algorithm. The method they introduce yields impressive results given the constraints imposed on the learning agent. However, the performance is still lower than that of a regular DQN agent learning without those constraints (see Figure 12 in [1]). Therefore, we recommend using our approach when batch updates and a replay buffer are available.
>
> Importantly, we demonstrate that our approach can be combined with Stream Q($\lambda$) [1] to further boost the learning speed. Indeed, in Section $5.5$ of the revised submission, we perform an additional study showing that the iterated Shared version of Stream Q($\lambda$), i.e., iS-Stream Q($\lambda$), increases the performance of the target-free version. This demonstrates that the presented approach is also relevant in the streaming scenario, where batch updates and a replay buffer are unavailable.
>
> [1] Elsayed, Mohamed, Gautham Vasan, and A. Rupam Mahmood. "Streaming deep reinforcement learning finally works." arXiv preprint arXiv:2410.14606 (2024).

---

> > ### Comment · Reviewer_Hr47 · 2025-11-27
> >
> > Thank for you addressing my concerns, I have updated my score accordingly.

---

> > > ### Author Response · Authors · 2025-12-01
> > >
> > > We are glad to see that the Reviewer appreciates our response. We thank the Reviewer for raising their score to $6$.

---

### Official Review · Reviewer_siCu · 2025-11-03

**Soundness:** 3
**Presentation:** 3
**Contribution:** 3
**Rating:** 8
**Confidence:** 3

**Summary:**

Proposes iterated shared Q-learning (iS-QL), which combines iterated Q-learning (prior work) with the novel approach of "Shared Features" for the target network. "Shared Features" replaces the standard target network with a single linear layer that shares its input features with the final linear layer of the online Q-network. Similar to a standard target network, the new target network head is a delayed copy of the online network head's parameters. This greatly reduces the memory required for training, since there is no longer a full target network, yet mostly retains the improved stability of a standard target network.

**Strengths:**

Experimental results suggest that iS-QL indeed increases return compared to target-free algorithms, but with significantly less memory usage than target-based algorithms.

The Fig 7 results on an LLM show iS-QL getting even higher return than the target-based algorithm, which is extra promising because that experiment is, among the experiments in the paper, debatably the closest to a real-world problem.

Large variety of experiments (online, offline, continuous control, and language-based RL)

The hyperparameter tuning protocol is clear (except for some things mentioned in "Weaknesses" of this review)

Code is included in the supplementary materials, and at a glance it looks fairly comprehensive

**Weaknesses:**

Introduces a hyperparameter, $K$, the number of Bellman iterations to learn in parallel. It appears to be mildly sensitive. Especially since, if I understand right, Fig 6 introduces an additional discounting hyperparameter for the iS-SAC $K = 9$ case.

As made clear in the paper (which is a strength), sometimes iS-QL does not match the returns of standard target-based algorithms.

It should be made clearer whether the target-based approach is given LayerNorm in continuous-action settings, or no normalization at all. If it is given no normalization at all, that would be unfair since iS-QL is given BatchNorm (which the paper notes does not help the target-based approach).

Fig 7 shows 600k gradient steps, but the appendix says 800k. Further, neither value is explained in the paper, and I don't think the original ILQL paper used either of those values.

Results use 5 or 10 seeds per task (but the large variety of experiments greatly mitigates this overall).

**Questions:**

> mitigate the brittleness of semi-gradient approaches and stabilize learning

Is this referring to two issues, or one issue?

&nbsp;

> Our proposed method, iterated Shared Q-Learning (iS-QL), bridges the performance gap between target-free and target-based approaches across various problems, while using a single Q-network, thus being a step forward towards resource-efficient reinforcement learning algorithms

You might split this sentence into two, and also say "stepping towards" instead of "thus being a step forward towards"

&nbsp;

> However, replacing look-up tables with non-linear function approximators and allowing off-policy samples to make the method more tractable

You might specify "tractable" more precisely here

&nbsp;

> This ultimately limits the size of the online network due to the constrained Video Random Access Memory (VRAM) of GPUs.

Memory limits also apply to, for example, TPUs

&nbsp;

> We propose storing only the smallest possible part of the target network, i.e., the parameters of the last linear layer,

This is not the "smallest possible part of the target network", because you could for example lag only half of the last linear layer. Even though I would guess that would not work well

&nbsp;

> Iterated Q-Network (Vincent et al., 2025)

"$Q$-network" and "Q-network" are used inconsistently. It would be better to stick with one

&nbsp;

> The optimal policy of a Markov Decision Process (MDP) with a discrete action space can be obtained by selecting for each state, the action that maximizes the optimal action-value function $Q^*$

This applies to continuous-action MDPs too

&nbsp;

> This is why Mnih et al. (2015) approximate the optimal action-value function with a neural network $Q_\theta$, represented by a vector of parameters $\theta$.

This confused me for a second, since I misread it as trying to explain why they use a neural network in particular

&nbsp;

> leveraging the contraction property of the Bellman operator $\Gamma$ to guide the optimization process toward the operator’s fixed point, i.e., the optimal action-value function $Q^*$.

The Bellman operator is often not a contraction (even when it can provably converge to $Q^*$). See for example Section 11.4.3 "Comparison between the two convergence proof techniques" in https://sites.google.com/view/rlfoundations/home

&nbsp;

> Where $\gamma$ is the discount factor linked to the MDP of interest.

"the discount factor linked to the MDP of interest" is a bit unclear, particularly since the discount factor used for training should often ideally be smaller than the discount factor used for evaluation. https://www.ijcai.org/Proceedings/16/Papers/626.pdf

&nbsp;

> Gallici et al. (2025) also develop a method for a streaming scenario, in which they rely on parallel environments to cope with the non-stationarity of the sample distribution

Would it be fair to say they also rely on LayerNorm for this?

&nbsp;

> gradient w.r.t. the

Is there a missing backslash in the LaTeX there? The space after the last "." looks a bit large

&nbsp;

> Importantly, we incorporate the insights provided by Gallici et al. (2025) to use LayerNorm (Ba et al., 2016) for the experiments with discrete action spaces, as we found it beneficial, even for the target-based approach. Similarly, we use BatchNorm (Ioffe & Szegedy, 2015), as suggested by Bhatt et al. (2024), to improve sample-efficiency in continuous action settings

This seems to imply BatchNorm helped more than LayerNorm for continuous-action settings? If so, it might help to make that even clearer.

&nbsp;

---

> ### Author Response · Authors · 2025-11-26
> **Answers to Weaknesses**
>
> We thank the Reviewer for the time spent reviewing the submission. We appreciate the extensive feedback and incorporated it in the revised submission.
>
> ## Weaknesses
>
> > I. Introduces a hyperparameter, $K$, the number of Bellman iterations to learn in parallel. It appears to be mildly sensitive. [...]
>
> While we agree that the performance of the presented approach can vary when changing the number of heads, we observe that the performance increases when considering more heads until a peak in performance is reached. Therefore, in Section $6$ of the revised submission, we provide some tuning guidance, advising to increase this hyperparameter until the performance starts decreasing, as with the learning rate. Furthermore, we now provide an analysis of the behavior of the presented method when varying this hyperparameter in Section $D.3$.
>
> > II. As made clear in the paper (which is a strength), sometimes iS-QL does not match the returns of standard target-based algorithms.
>
> The main objective of this work is to present a method to improve the learning speed of target-free algorithms while maintaining a low memory footprint. We used the target-based version as an upper bound on the performance. By sharing features and enriching the loss with iterated Q-learning, we observe significant improvements, even outperforming the target-based version in some settings.
>
> > III. It should be made clearer whether the target-based approach is given LayerNorm in continuous-action settings. [...]
>
> This comment relates to the experiment done with the SimbaV2 architecture, one of the most advanced architectures for training a SAC agent. All algorithms presented in this experiment, including the target-based approach, use $\ell_2$-norm, which is an alternative to layer norm. We favored $\ell_2$-norm over layer normalization so that the target-based approach is identical to the published version of SimbaV2. We clarified this point in the revised version of our submission in the legend of Figure $25$.
>
> > IV. Fig $7$ shows $600k$ gradient steps, but the appendix says $800k$. Further, neither value is explained in the paper. [...]
>
> We thank the Reviewer for pointing out this typo; the correct value is $600\ 000$ gradient steps. We corrected the value in Table $3$. The value $600\ 000$ was chosen by running TB-ILQL until the performance reported in Table $6$ of the original ILQL paper was reached. We added this explanation in Paragraph "Wordle setup" in Section $A$.
>
> > V. Results use $5$ or $10$ seeds per task (but the large variety of experiments greatly mitigates this overall).
>
> We used $5$ seeds for all Atari environments and for the Wordle experiment, and $10$ seeds for the experiments on the DeepMind Control Suite, as indicated in Section $5$. We argue that these numbers are standard and often used in the literature.

---

> > ### Author Response · Authors · 2025-11-26
> > **Answers to Questions**
> >
> > ## Questions
> >
> > > 1. "mitigate the brittleness of semi-gradient approaches and stabilize learning"
> > > Is this referring to two issues, or one issue?
> >
> > The latter is a consequence of the former. Indeed, mitigating the brittleness of semi-gradient approaches stabilizes the learning process.
> >
> > > 2. [...] You might split this sentence into two, and also say "stepping towards" instead of "thus being a step forward towards"
> >
> > We thank the Reviewer for the suggestion. We modified the last sentence of the abstract by replacing "a step forward towards" with "stepping towards".
> >
> > > 3. "However, replacing look-up tables with non-linear function approximators and allowing off-policy samples to make the method more tractable"
> > > You might specify "tractable" more precisely here
> >
> > We replaced "tractable" with "scalable".
> >
> > > 4. [...] Memory limits also apply to, for example, TPUs
> >
> > We incorporated these suggestions in the revised submission.
> >
> > > 5. [...] This is not the "smallest possible part of the target network", because you could for example lag only half of the last linear layer. Even though I would guess that would not work well.
> >
> > In the revised version, we removed this part of the sentence to avoid confusion.
> >
> > > 6. "$Q$-network" and "Q-network" are used inconsistently.
> >
> > In the revised version, we ensured that "$Q$-function", "$Q$-learning", and "$Q$-network" are consistently written.
> >
> > > 7. "The optimal policy of a Markov Decision Process (MDP) with a discrete action space can be obtained by selecting for each state, the action that maximizes the optimal action-value function $Q^*$"
> > > This applies to continuous-action MDPs too
> >
> > We removed this part of the sentence in the revised submission to refer to the general case.
> >
> > > 8. "This is why Mnih et al. (2015) approximate the optimal action-value function with a neural network $Q_{\theta}$, represented by a vector of parameters $\theta$."
> > > This confused me for a second, since I misread it as trying to explain why they use a neural network in particular
> >
> > We modified the sentence to improve the reading flow.
> >
> > > 9. "leveraging the contraction property of the Bellman operator $\Gamma$ to guide the optimization process toward the operator’s fixed point, i.e., the optimal action-value function."
> > > The Bellman operator is often not a contraction. [...]
> >
> > We thank the Reviewer for the suggestion. In this sentence, we refer to the contraction property of the Bellman operator in the value space. We included this precision in the revised submission.
> >
> > > 10. "the discount factor linked to the MDP of interest" is a bit unclear, particularly since the discount factor used for training should often ideally be smaller than the discount factor used for evaluation. [...]
> >
> > We thank the reviewer for the suggestion. We removed this part of the sentence to avoid confusion.
> >
> > > 11. "Gallici et al. (2025) also develop a method for a streaming scenario, in which they rely on parallel environments to cope with the non-stationarity of the sample distribution"
> > > Would it be fair to say they also rely on LayerNorm for this?
> >
> > While we agree that layer norm helps the optimization process, we argue that the use of parallel environments is the main technique used to overcome the non-stationarity of the sample distribution, as it allows the agent to have access to different parts of the state space at each gradient step.
> >
> > > 12. "gradient w.r.t. the"
> > > Is there a missing backslash in the LaTeX there? [...]
> >
> > There is indeed a missing blackslash. We corrected it in the revised submission.
> >
> > > 13. [...] This seems to imply BatchNorm helped more than LayerNorm for continuous-action settings?
> >
> > In the continuous action setting, we used the SimbaV2 architecture, which incorporates $\ell_2$-normalization layers. The $\ell_2$-norm is a substitute for layer normalization. We found that BatchNorm further helped in improving sample efficiency in this setting, except for the target-based approach, as shown in Figure $25$. However, we did not study which normalization technique is most impactful, as we believe this would not fit the scope of this work.

---

> > > ### Comment · Reviewer_siCu · 2025-11-26
> > >
> > > My concerns have largely been addressed, and I maintain my score of 8. Thank you for improving the paper.
> > >
> > > Just to be clear, I still dislike the additional $0.25$ "discounting factor" (which perhaps should have a different name) used for Fig 6, which was not addressed in the rebuttal, but does not change my rating
> > >
> > > > serves as the goal standard
> > >
> > > in the paper should be "gold standard"
> > >
> > > >  We argue that [using 5 - 10 seeds is] standard and often used in the literature.
> > >
> > > This is true, though "standard" and "often used" does not mean "good". However, as I noted before, the large variety of experiments greatly mitigates this weakness.

---

> > > > ### Author Response · Authors · 2025-12-01
> > > >
> > > > We are happy that the Reviewer's concerns "have largely been addressed" and that the Reviewer confirms their positive evaluation of our work.
> > > >
> > > > > in the paper should be "gold standard"
> > > >
> > > > We corrected this typo in the revised submission.
> > > >
> > > > > I still dislike the additional $0.25$ "discounting factor" used for Fig 6, [...], but does not change my rating
> > > >
> > > > We thank the Reviewer for the additional comment. We first stress that, in this experiment, iS-SAC $K=1$ bridges the performance gap between the target-free and target-based approaches, which is the main goal of the work. Then, as mentioned in Line $401$, we "find it beneficial to give more importance to the first Bellman updates by scaling the future terms in the loss by a discounting factor of $0.25$". Finally, in the revised submission, we added a clarification, stating that: "we leave the investigation of finding the best way to weight each term in the loss to future work. In Section D.4, we provide a first direction leveraging the concept of meta-gradient reinforcement learning (Xu et al., 2018) to tune learnable coefficients assigned to each term in the loss during training".

---

### Author Response · Authors · 2025-11-26
**Additions to the revised submission**

We thank the Reviewers for their valuable feedback, which helped us to improve the submission. We have submitted a revision with the following additions highlighted in blue:

- We evaluate the presented approach in a streaming RL scenario, in Section $5.5$. We show that it boosts the learning speed of the target-free approach. ```Asked by Hr47 and Bpoz```

- We perform an ablation study on the number of Bellman updates $K$, in Section $D.3$. We show that iS-DQN's performance remains stable for various values of $K$, outperforming the target-free approach in all cases. ```Asked by siCu, 959t, and Bpoz```

- We evaluate the presented approach when the MellowMax operator is used instead of the maximum operator. We show that the performance gap is recovered by the presented approach. ```Asked by 959t```

- We analyse the behavior of iS-DQN when varying the target update period $T$. iS-DQN outperforms both the target-free and target-based variants across all target update periods considered in the study. ```Asked by 959t```

- We introduce a method to tune the importance given to each Bellman update adaptively. We show that meta-gradient naturally provides a way to achieve that goal. ```Asked by Bpoz```

- We perform a pilot study showing that combining mixed precision training with iS-DQN leads to promising results, while further reducing the resource utilization. ```Asked by 959t```

---

### Meta-Review · Area_Chair_9uiu · 2025-12-21

**Summary:**

Most important shared concerns

* Positioning vs prior/parallel work (esp. streaming RL papers) and whether comparisons are sufficient (strongest for Hr47; echoed indirectly via “direct quantitative comparisons” in 959t).
* Lack of formal theory (convergence/stability analysis) despite stability claims (Hr47 + 959t).
* The sensitivity of the method to Hyperparameter 𝐾 choice
* Practical compute story: memory improves, but compute/wall-clock speed not necessarily improved; want clearer practical implications (959t, Bpoz).
* Clarity / experimental-detail correctness (normalization fairness, step-count mismatch, naming/notation clarity) (siCu, 959t).

**Reviewer Concerns:**

Most clearly addressed

* The hyperparameter tuning concern: the authors added tuning guidance + ablations (appears to satisfy siCu, and was directly answered to 959t/Bpoz).
* The streaming RL positioning: added explicit argument + new streaming experiment combining with Stream Q, which directly targets Hr47’s main objection.
* Clarity/correctness fixes: normalization clarification, typo corrections, step-count correction, naming consistency (strongly for siCu; some for 959t).

Most outstanding (or only partially addressed)

* Formal theory remains absent (authors consistently substitute empirical/mechanistic analyses).
* The wall-clock compute story remains a framing argument (sample efficiency matters when real-world env steps dominate) rather than a direct demonstration of time-to-performance improvements on real hardware setups.
* The “direct quantitative comparisons under identical settings” request is partially addressed (targeted MellowMax study + incorporating norms), but not obviously a comprehensive apples-to-apples suite vs each cited alternative.

**Reviewer Scores:**

siCu: explicitly indicated no change (stays at 8) and says concerns largely addressed, with one lingering dislike that doesn’t affect rating.

Hr47: explicitly increased score after authors addressed the Elsayed/Vasan comparison concern (exact numbers not visible).

959t & Bpoz: had no stated change because they did not engage during the discussion, they might have been willing to increase scores.

---

### Decision · Program_Chairs · 2026-01-26

Accept (Poster)